# CA2 neuronal activity controls hippocampal low gamma and ripple oscillations

**Georgia M Alexander, Logan Y Brown, Shannon Farris, Daniel Lustberg[†], Caroline Pantazis[‡], Bernd Gloss, Nicholas W Plummer, Patricia Jensen, Serena M Dudek***

Neurobiology Laboratory, National Institute of Environmental Health Sciences, National Institutes of Health, North Carolina, United States

**\*For correspondence:**
dudek@niehs.nih.gov

**Present address:** [†]Department of Human Genetics, Emory University School of Medicine, Georgia, United States; [‡]Brain Health Institute, Rutgers University, New Jersey, United States

**Competing interests:** The authors declare that no competing interests exist.

**Abstract** Hippocampal oscillations arise from coordinated activity among distinct populations of neurons and are associated with cognitive functions. Much progress has been made toward identifying the contribution of specific neuronal populations in hippocampal oscillations, but less is known about the role of hippocampal area CA2, which is thought to support social memory. Furthermore, the little evidence on the role of CA2 in oscillations has yielded conflicting conclusions. Therefore, we sought to identify the contribution of CA2 to oscillations using a controlled experimental system. We used excitatory and inhibitory DREADDs to manipulate CA2 neuronal activity and studied resulting hippocampal-prefrontal cortical network oscillations. We found that modification of CA2 activity bidirectionally regulated hippocampal and prefrontal cortical low-gamma oscillations and inversely modulated hippocampal ripple oscillations in mice. These findings support a role for CA2 in low-gamma generation and ripple modulation within the hippocampus and underscore the importance of CA2 in extrahippocampal oscillations.
DOI: https://doi.org/10.7554/eLife.38052.001

## Introduction

Area CA2 has become appreciated as a distinct subfield of the hippocampus based on several molecular, synaptic, anatomical, and functional properties (see *Dudek et al., 2016* for review). We and others have recently identified similarities and differences between CA2 and the neighboring CA1 and CA3 subfields based on action potential firing *in vivo* (*Alexander et al., 2016*; *Mankin et al., 2015*; *Lee et al., 2015*; *Lu et al., 2015*; *Kay et al., 2016*). In addition to action potential firing, another form of neuronal communication may be achieved through synchronized oscillations, which reflect the summated electrical activity of a population of neurons and can be detected in local field potentials (LFPs). CA1 and CA3 networks propagate oscillations in three primary frequency bands: theta (~5 – 10 Hz), gamma (~30 – 100 Hz) and sharp-wave ripples (~100 – 250 Hz). A few studies have reported properties of network oscillations in CA2 (*Kay et al., 2016*; *Oliva et al., 2016*; *Boehringer et al., 2017*), but none of them have examined CA2 gamma oscillations or the impact of CA2 oscillations on extrahippocampal structures.

In the hippocampus, high- and low-gamma oscillations are thought to arise from two distinct sources and likely play separate roles in memory (*Colgin et al., 2009*). High-gamma (~60 – 100 Hz) oscillations in CA1 are prevalent in *stratum lacunosum-moleculare* (*Schomburg et al., 2014*), co-occur with high-gamma oscillations in medial entorhinal cortex (MEC) (*Colgin et al., 2009*), are increased by Gq-Designer Receptors Exclusively Activated by Designer Drugs (DREADD)-mediated activation of cortex (*Alexander et al., 2009*), and are impaired by lesioning of EC (*Bragin et al., 1995*), leading to the conclusion that high-gamma oscillations arise from MEC. High-gamma is

thought to contribute to memory encoding because high-gamma power is increased upon exploration of novel stimuli (*Zheng et al., 2016*; *Kemere et al., 2013*). Low-gamma (~30 – 55 Hz) oscillations in CA1 are prevalent in *stratum radiatum* (*Schomburg et al., 2014*), synchronize with low-gamma in CA3 (*Colgin, 2015*), and become more evident upon EC lesioning (*Bragin et al., 1995*), supporting the conclusion that low-gamma oscillations arise from CA3. Low-gamma oscillations are believed to promote memory retrieval because the magnitude of low-gamma coupling to theta oscillations correlates with performance on learned behavioral tasks (*Tort et al., 2009*; *Shirvalkar et al., 2010*). Interestingly, complete silencing of the synaptic output of CA3 with tetanus toxin light chain does not completely impair low-gamma oscillations (*Middleton and McHugh, 2016*), suggesting the presence of another source of low-gamma oscillations.

Another prominent oscillation seen in hippocampus is sharp-wave ripple oscillations, which are high frequency (~100 – 250 Hz), short-duration electrical events prominently seen in LFP recordings from CA1 during awake immobility and slow wave sleep (*Buzsáki et al., 1992*). Sharp waves are thought to arise from the synchronous firing of CA3 pyramidal cells, which depolarize the apical dendrites of CA1 pyramidal cells. The synchronous CA3 firing recruits excitatory and inhibitory neurons in CA1 to generate ripples (*Buzsáki et al., 1992*; *Ylinen et al., 1995*). A role for CA2 neurons in sharp-wave ripples has recently been suggested based on three *in vivo* electrophysiology studies (*Kay et al., 2016*; *Oliva et al., 2016*; *Boehringer et al., 2017*), although consensus has not been reached on the precise role that these neurons play. found that CA2 is the only hippocampal subregion to have a substantial population of neurons that cease firing during ripples (termed 'N cells'), whereas nearly all pyramidal cells queried in neighboring subfields fired during ripples. Although not associated with ripples, these N cells fired at high rates during low running speed or immobility (*Kay et al., 2016*). *Oliva et al. (2016)* later reported that CA2 pyramidal cell activity ramps up before the onset of sharp-wave ripples, leading these authors to conclude that CA2 neurons play a leading role in ripple generation. By contrast, *Boehringer et al. (2017)* later found that chronic silencing of CA2 pyramidal cell output leads to the occurrence of epileptic discharges arising from CA3, which the authors suggested reflect anomalous ripple oscillations. Accordingly, findings of the Boehringer study do not appear to support the conclusion of that CA2 neurons initiate ripples. Given the disparate conclusions of these reports, further study is required to clarify the role of CA2 neuronal activity in ripple generation.

Area CA2 has recently been recognized for its role in processing long-term memories containing socially relevant information in rodents (*Alexander et al., 2016*; *Hitti and Siegelbaum, 2014*; *Pagani et al., 2015*; *Smith et al., 2016*). Interestingly, a mouse model of schizophrenia that shows hypoactive CA2 pyramidal cells *in vitro* also shows impaired social behavior (*Piskorowski et al., 2016*). Further, long-range synchrony between hippocampus and prefrontal cortex (PFC), including low-gamma coherence, is impaired in another mouse model of schizophrenia (*Sigurdsson et al., 2010*), raising the question of how altering CA2 pyramidal cell activity experimentally may impact social behavior and synchrony between hippocampus and PFC.

In this study, we present evidence that selective, acute activation or inhibition of CA2 pyramidal cells using Cre-dependent expression of Gq- and Gi-coupled DREADD receptors (hM3Dq and hM4Di, respectively; *Alexander et al., 2009*; *Armbruster et al., 2007*) bidirectionally modulates low-gamma oscillations in both hippocampus and PFC and ripple occurrence in hippocampus.

## Results

### Increasing CA2 pyramidal cell activity increases hippocampal and prefrontal cortical low-gamma power

To gain selective genetic access to molecularly-defined CA2, we generated a tamoxifen-inducible mouse line, *Amigo2*-icreERT2. When crossed with a Cre-dependent tdTomato reporter mouse line (*Madisen et al., 2010*), we observed robust expression of tdTomato in CA2 of brain sections from *Amigo2*-icreERT2+; *ROSA*-tdTomato± mice treated with tamoxifen (*Figure 1A–E*). Expression of tdTomato colocalized with the CA2 pyramidal cell marker, PCP4 (*Kohara et al., 2014*), in 91.4% of neurons (N = 6 mice, *Figure 1A,B,E*), and tdTomato colocalized with a marker of hippocampal pyramidal neurons (N = 6; *Figure 1C*, *Figure 1—figure supplement 1A*) but not inhibitory neurons (N = 3; *Figure 1D*, *Figure 1—figure supplement 1B*). Expression of tdTomato was also observed in

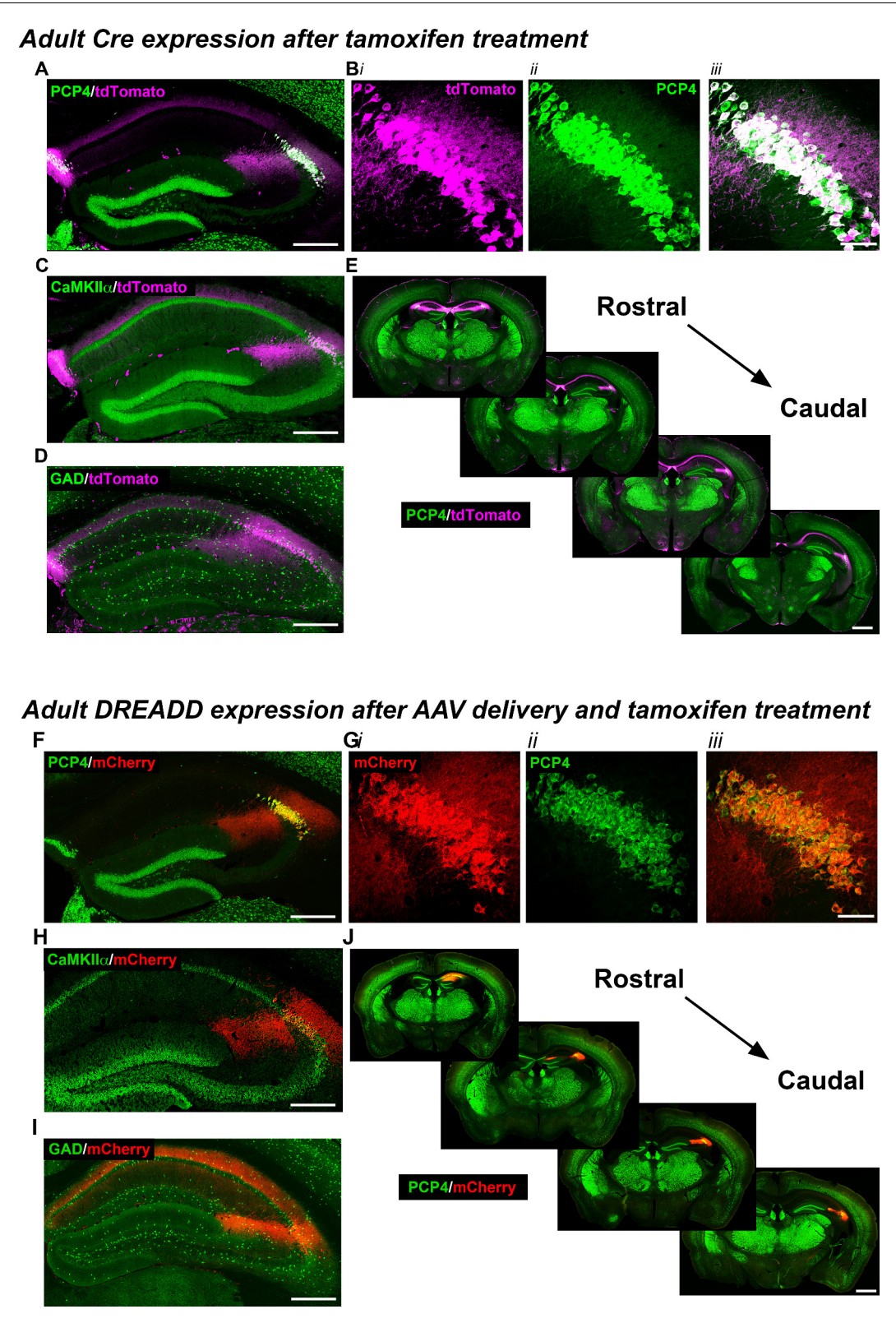

## Adult Cre expression after tamoxifen treatment

## Adult DREADD expression after AAV delivery and tamoxifen treatment

**Figure 1.** Expression of the Cre indicator, tdTomato, in *Amigo2*-icreERT2+; *ROSA*-tdTomato± mice (A–E) or mCherry-tagged DREADD receptors (F–J) in *Amigo2*-icreERT2 mice. (A–B) Co-expression of tdTomato and PCP4, a marker for CA2 neurons, in the coronal hippocampal section (A) or CA2 only (B). In B, *i* shows tdTomato expression, *ii* shows PCP4 expression and *iii* shows the merged image. (C) Co-expression of tdTomato and CaMKIIα, a marker for principal neurons in hippocampus (see also *Figure 1—figure supplement 1A*). (D) Expression of tdTomato and GAD, a marker for inhibitory

*Figure 1 continued*

neurons. Cre-dependent tdTomato expression did not colocalize with GAD (see also *Figure 1—figure supplement 1B*). (E) Expression of tdTomato colocalizes with expression of PCP4 across the rostral to caudal extent of CA2. (F–J) Coronal sections from *Amigo2*-icreERT2+ mice infused unilaterally with AAV-hSyn-DIO-hM3D(Gq)-mCherry (hM3Dq AAV; F–H,J) or bilaterally with AAV-hSyn-DIO-hM4D(Gi)-mCherry (hM4Di AAV; I) and treated with tamoxifen. (F–G) Expression of hM3Dq-mCherry and the CA2-specific marker PCP4, in the hippocampus (F) and CA2 (G). In (G), *i* shows DREADD-mCherry expression, *ii* shows PCP4 expression and *iii* shows the merged image. Expression of DREADD-mCherry colocalizes with CaMKIIα (H) but does not colocalize with GAD (I; see also *Figure 1—figure supplement 1C–D*). Note that hM4Di-mCherry (shown in I) fills axons projecting to CA1. (J) Expression of hM3Dq-mCherry colocalizes with expression of PCP4 across the rostral to caudal extent of CA2. Scale bars = 200 µm (A, C, D, F, H, I), 50 µm (B, G) and 1 mm (E, J). See also *Figure 1—figure supplement 1* and *Figure 1—figure supplement 2*.
DOI: https://doi.org/10.7554/eLife.38052.002
The following figure supplements are available for figure 1:

**Figure supplement 1.** High-magnification co-expression images of tdTomato (A–B) or mCherry (C–D) with markers of excitatory and inhibitory neurons.
DOI: https://doi.org/10.7554/eLife.38052.003
**Figure supplement 2.** High-magnification co-expression images of tdTomato (A) or mCherry (B) with PCP4, a marker for CA2 neurons in negative control animals.
DOI: https://doi.org/10.7554/eLife.38052.004

fasciola cinerea, extra-hippocampal brain structures, and associated with vasculature. In control experiments, *Amigo2*-icreERT2+; *ROSA*-tdTomato± animals treated with corn oil (the tamoxifen vehicle) showed no tdTomato expression (N = 3; *Figure 1—figure supplement 2A*).

Infusion of AAVs encoding Cre-dependent hM3Dq (*Figure 1F–H,J*) or hM4Di (*Figure 1I*) with the neuron-specific human synapsin promoter into *Amigo2*-icreERT2+ mice allowed for selective expression of mCherry-DREADD in CA2 pyramidal neurons without expression in fasciola cinerea, outside of the hippocampus, or in the vasculature, as detected by co-expression of mCherry with PCP4 (N = 4; *Figure 1F–G,J*). Expression of mCherry also colocalized with the pyramidal cell marker, CaMKIIα (N = 4; *Figure 1H*, *Figure 1—figure supplement 1C*), but not the interneuron marker, glutamic acid decarboxylase (GAD), in *GAD*-eGFP+; *Amigo2*-icreERT2+ mice (N = 4; *Figure 1I*, *Figure 1—figure supplement 1D*). In control *Amigo2*-icreERT2- mice infused with hM3Dq AAV, mCherry expression was absent (N = 4; *Figure 1—figure supplement 2B*).

With genetic access to CA2 pyramidal cells gained, we could selectively modify activity of CA2 neurons *in vivo* with excitatory or inhibitory DREADDs and measure the resulting network and behavioral effects. One advantage of DREADDs is that compared with tetanus toxin light chain, which permanently silences neuronal output, DREADDs permit transient modification of neuronal activity (between 4 and 24 hr; see also *Figure 5—figure supplement 1*), reducing the potential for compensatory circuit reorganization.

To measure the effect of increasing CA2 neuronal activity on hippocampal and prefrontal cortical population oscillatory activity, *Amigo2*-icreERT2+ and control *Amigo2*-icreERT2- mice were infused unilaterally with hM3Dq AAV, treated with tamoxifen and then implanted with electrodes in hippocampus and PFC (see *Figure 2—figure supplement 1*) to measure changes in LFP. To confirm that hM3Dq increased neuronal activity, single-unit firing rate was measured from CA2/proximal CA1 pyramidal neurons. Clozapine-N-oxide (CNO) treatment dose-dependently increased the firing rate of pyramidal neurons following CNO administration (*Figure 2—figure supplement 2*).

Next, *Amigo2*-icreERT2+ and control *Amigo2*-icreERT2- mice were treated with various doses of CNO or vehicle as control, and hippocampal LFPs were assessed for CNO treatment-dependent effects using spectral analyses, focusing on theta (5 – 10 Hz), low-gamma (30 – 60 Hz) and high-gamma (65 – 100 Hz) oscillations. We measured oscillatory power during the 30 to 60 min time window following treatment during each of running and resting behavioral periods (*Figure 2*; see also *Figure 2—figure supplement 3*). We found a significant increase in low-gamma power following CNO administration during running for all doses tested (N = 8; $F_{(1.904, 13.33)}$ = 9.457, p = 0.0030, repeated-measures one-way ANOVA with Geisser-Greenhouse correction for unequal variance; 0.5 mg/kg: p = 0.0286; 1 mg/kg: p = 0.0286; 2 mg/kg: p = 0.0286; 4 mg/kg: *p* = 0.0191, Holm-Sidak post hoc test for multiple comparisons versus vehicle; *Figure 2Biii*). To address whether change in locomotor activity could have produced the change in low-gamma power, we measured the percent of time spent running and the mean running speed following vehicle or drug treatment. We found no significant effect of treatment on locomotion (percent of time running: main effect of genotype: F

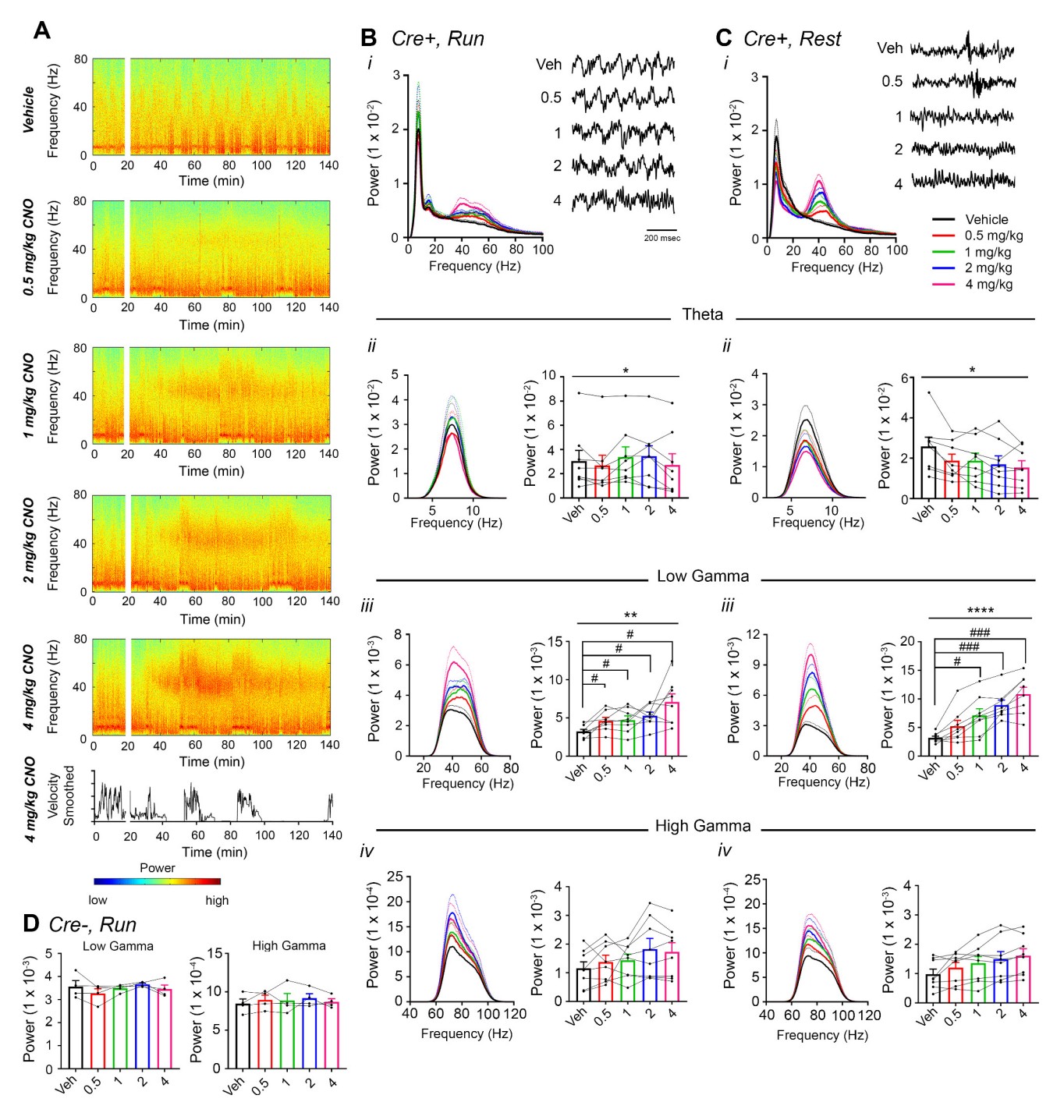

**Figure 2.** CNO treatment dose-dependently increases low-gamma power in hippocampus of hM3Dq-infused *Amigo2*-icreERT2+ mice. (**A**) Spectrograms of hippocampal LFP recordings depicting LFP power according to different frequencies over time. Vehicle/CNO administration time is shown by the white bar, and the treatment is shown to the left of each spectrogram. Locomotor velocity is shown in the bottom panel, corresponding to the 4 mg/kg CNO spectrogram. Note the increased theta band power during periods of running compared with periods of resting. (**B–C**) Power measures for hippocampal LFPs in *Amigo2*-icreERT2+ mice during periods of running (**B**) and resting (**C**). For each of B and C: (*i*) Power spectral density plots from LFPs for frequencies up to 100 Hz. (*ii-iv*) Power spectral density plots and peak power from LFPs filtered in the theta (5 – 10 Hz; *ii*), low-gamma (30 – 60 Hz; *iii*), or high-gamma (65 – 100 Hz; *iv*) frequency ranges. In B*i* and C*i*, raw LFP traces are shown to the right of the power spectral density plot for each treatment, sampled during a period of running (B*i*) or resting (C*i*) during the 30 – 60 min following treatment listed. Scale bar applies to both sets of traces. All recordings are taken from the same recording site in the same animal. In B*ii-iv* and C*ii-iv*, plots on the left show power spectral density plots for each frequency band, and plots on the right show mean peak power for the population of animals in colored bars and data

*Figure 2 continued on next page*

*Figure 2 continued*

from individual animals as black dots. (B*ii*) Theta power varied significantly upon treatment during running (N = 8 mice (three female, five male); Friedman statistic = 11.3; p = 0.0234, results of post hoc tests not significant). (B*iii*) CNO treatment produced a significant dose-dependent increase in low-gamma power during running ($F_{(1.904, 13.33)}$ = 9.457, p = 0.0030, repeated-measures one-way ANOVA with Geisser-Greenhouse correction for unequal variance; results of Holm-Sidak post hoc tests are shown by symbols. (B*iv*) CNO treatment did not significantly affect high-gamma power during running ($F_{(1.384, 9.69)}$ = 2.288, p = 0.1602, repeated-measures one-way ANOVA with Geisser-Greenhouse correction for unequal variance). (C*ii*) Theta power varied significantly upon treatment during rest (same N; $F_{(1.972, 13.81)}$ = 4.825, p = 0.0261, repeated-measures one-way ANOVA with Geisser-Greenhouse correction for unequal variance; results of post hoc tests not significant). (C*iii*) CNO treatment produced a significant dose-dependent increase in low-gamma power during rest ($F_{(2.306, 16.15)}$ = 32.2), p < 0.0001, repeated-measures one-way ANOVA with Geisser-Greenhouse correction for unequal variance; results of Holm-Sidak post hoc tests are shown by symbols). (C*iv*) CNO treatment did not significantly affect high-gamma power during rest ($F_{(1.286, 9.003)}$ = 4.775, p = 0.0501, repeated-measures one-way ANOVA with Geisser-Greenhouse correction for unequal variance). (D) Peak low-gamma (left plot) and high-gamma (right plot) power for the population of *Amigo2*-icreERT2- mice infused with hM3Dq, treated with tamoxifen and challenged with CNO. Neither low-gamma power nor high-gamma power changed significantly in response to CNO administration during running (N = 4 male mice; low-gamma: $F_{(1.669, 5.006)}$ = 1.36, p = 0.3281; high-gamma: $F_{(1.895, 5.684)}$ = 0.5079, p = 0.6175, repeated-measures one-way ANOVA with Geisser-Greenhouse correction for unequal variance). *p < 0.05, **p < 0.01, ****p < 0.0001, one-way ANOVA; #p < 0.05, ###p < 0.001, Holm-Sidak *post hoc* test. See also *Figure 2—figure supplements 1–8*.

DOI: https://doi.org/10.7554/eLife.38052.005

The following figure supplements are available for figure 2:

**Figure supplement 1.** Electrode localization for Cre+ and Cre- *Amigo2*-icreERT2 mice infused with hM3Dq AAV and implanted with electrodes into hippocampus and PFC.
DOI: https://doi.org/10.7554/eLife.38052.006

**Figure supplement 2.** CNO dose-dependently increased firing rate of hippocampal pyramidal neurons in an hM3Dq-expressing mouse.
DOI: https://doi.org/10.7554/eLife.38052.007

**Figure supplement 3.** High-gamma power and theta power were significantly greater during periods of running than periods of resting in hM3Dq animals following vehicle treatment, as measured from non-normalized LFPs.
DOI: https://doi.org/10.7554/eLife.38052.008

**Figure supplement 4.** Percent of time running (A, C, E) and mean running speed during periods of run (B, D, F) for hM3Dq (A, B) and hM4Di (C, D) animals for each treatment.
DOI: https://doi.org/10.7554/eLife.38052.009

**Figure supplement 5.** Change in gamma power by hM3Dq is selective for low-gamma and does not significantly affect highgamma during either run (A) or rest (B).
DOI: https://doi.org/10.7554/eLife.38052.010

**Figure supplement 6.** The magnitude of hM3Dq-mediated change in low-gamma power did not vary significantly as a function of proximal or distal recording location for periods of running (A–B) or rest (C–D).
DOI: https://doi.org/10.7554/eLife.38052.011

**Figure supplement 7.** Comodulation of low-gamma amplitude and theta phase in hM3Dq mice with vehicle or CNO treatment.
DOI: https://doi.org/10.7554/eLife.38052.012

**Figure supplement 8.** CNO did not affect low-gamma power in hippocampus of hM3Dq-infused Cre- *Amigo2*-icreERT2 mice.
DOI: https://doi.org/10.7554/eLife.38052.013

$F_{(1, 10)}$ = 0.1317, *p* = 0.7242; main effect of treatment: $F_{(4, 40)}$ = 0.5251, *p* = 0.7178; interaction: $F_{(4, 40)}$ = 1.265, *p* = 0.2996; running velocity: main effect of genotype: $F_{(1, 10)}$ = 0.2178, p = 0.6507); main effect of treatment: $F_{(4, 40)}$ = 0.1857; p = 0.9445); interaction: $F_{(4, 40)}$ = 0.8774, p = 0.4860; two-way ANOVA; *Figure 2—figure supplement 4A–B*). Further, during periods of rest, we also found a significant increase in low-gamma power following CNO administration ($F_{(2.306, 16.15)}$ = 32.2, p < 0.0001, repeated-measures one-way ANOVA with Geisser-Greenhouse correction for unequal variance; 0.5 mg/kg: p = 0.1008; 1 mg/kg: p = 0.0161; 2 mg/kg: p = 0.0002; 4 mg/kg: p = 0.0004, Holm-Sidak post hoc test for multiple comparisons versus vehicle; *Figure 2Ciii*).

High-gamma power was not significantly changed by CNO treatment compared with vehicle during either run ($F_{(1.384, 9.69)}$ = 2.288, p = 0.1602, repeated-measures one-way ANOVA with Geisser-Greenhouse correction for unequal variance, *Figure 2Biv*) or rest ($F_{(1.286, 9.003)}$ = 4.775, p < 0.0501, repeated-measures one-way ANOVA with Geisser-Greenhouse correction for unequal variance, *Figure 2Civ*). To assess whether gamma power changes were specific to the low-gamma range, we also compared the change in gamma power for low- and high-gamma together using a two-way ANOVA. This analysis confirmed the selective increase in low-gamma power during running (main effect of treatment: $F_{(4, 28)}$ = 9.605, p < 0.0001; main effect of gamma range: $F_{(1, 7)}$ = 111.1, p < 0.0001; interaction: $F_{(4, 28)}$ = 6.945, p = 0.0005, two-way ANOVA; Bonferroni post-hoc tests:

low-gamma: 0.5 mg/kg, p = 0.0204; 1 mg/kg, p = 0.0112; 2 mg/kg, p = 0.0004; 4 mg/kg, p < 0.0001; high-gamma: 0.5 and 1 mg/kg, p > 0.9999; 2 mg/kg, p = 0.5857; 4 mg/kg, p = 0.8531; *Figure 2—figure supplement 5A*). We also confirmed the selective increase in low-gamma power during periods of rest (main effect of treatment: F(4, 28) = 38.24, p < 0.0001; main effect of gamma range: F(1,7) = 51.01, p = 0.0002; interaction: F(4, 28) = 24.54, p < 0.0001, two-way ANOVA; Bonferroni post-hoc tests: low gamma: 0.5 mg/kg, p = 0.0105; 1 mg/kg, p < 0.0001; 2 mg/kg, p < 0.0001; 4 mg/kg, p < 0.0001; high gamma: all doses, p > 0.9999; *Figure 2—figure supplement 5B*).

Post hoc assessment of electrode position showed that recordings from wires targeted toward CA2/proximal CA1 resulted in CA2 placement in some animals and proximal to intermediate CA1 placement in other animals (*Figure 2—figure supplement 1*; order of images denotes proximal to distal electrode placements). We separated data from animals with recordings taken from CA2 (N = 2), proximal CA1 (N = 4) and more intermediate CA1 (N = 2) and then compared changes in low-gamma power for the three groups (*Figure 2—figure supplement 6*). We found no significant difference in the magnitude of low-gamma power change between recordings from close to CA2 and further from CA2 during periods of run (main effect of recording location: F(2, 5) = 0.6411, p = 0.5651; main effect of treatment: F(4, 20) = 6.668, p = 0.0014; interaction: F(8, 20) = 0.7554, p = 0.6443; *Figure 2—figure supplement 6A–B*) or periods of rest (main effect of location: F(2, 5) = 1.307, p = 0.3494; main effect of treatment: F(4, 20) = 29.98, p < 0.0001; interaction: F(8, 20) = 1.478, p = 2268; *Figure 2—figure supplement 6C–D*).

We also measured theta phase, low-gamma amplitude comodulation during periods of running and found that neither theta phase at which low-gamma amplitude was greatest nor the modulation index was significantly affected by CNO treatment (N = 8, phase: F(1.266, 8.864) = 0.4696, p = 0.5552; modulation index: F(2.475, 17.33) = 3.11, p = 0.0612, repeated-measures one-way ANOVA with Geisser-Greenhouse correction for unequal variance; *Figure 2—figure supplement 7A*).

In contrast, in control *Amigo2*-icreERT2- mice, during periods of running, CNO treatment had no effect on low- or high-gamma power (N = 4; low gamma: F(1.669, 5.006) = 1.36, p = 0.3281; high-gamma: F(1.895, 5.684) = 0.5079, p = 0.6175, repeated-measures one-way ANOVA with Geisser-Greenhouse correction for unequal variance; *Figure 2D*, *Figure 2—figure supplement 5C–D*, *Figure 2—figure supplement 8*) or theta phase low-gamma amplitude comodulation (phase: F(1.449, 4.346) = 0.6923, p = 0.5033; modulation index: F(1.329, 3.988) = 0.3098, p = 0.6688, repeated-measures one-way ANOVA with Geisser-Greenhouse correction for unequal variance; *Figure 2—figure supplement 7B*). Further, analysis of low-gamma power in *Amigo2*-icreERT2+ and *Amigo2*-icreERT2- together yielded a significant interaction between treatment and genotype (F(4, 40) = 4.55, p = 0.0040, two-way ANOVA) with post hoc tests showing significant increases in low-gamma power for *Amigo2*-icreERT2+ animals and not *Amigo2*-icreERT2- animals (0.5 mg/kg: p = 0.0632, 1 mg/kg: p = 0.0385, 2 mg/kg: p = 0.0021, 4 mg/kg: p < 0.0001; Bonferonni's multiple comparisons test).

Given the role of the hippocampal-prefrontal cortical pathway in spatial working memory and the involvement of gamma synchrony between the two structures (*Hitti and Siegelbaum, 2014*) as well as the previous finding that gamma synchrony is impaired in a mouse model of schizophrenia (*Sigurdsson et al., 2010*), we wondered what contribution CA2 activity makes toward PFC gamma oscillations. Therefore, we asked whether hippocampal low-gamma oscillations resulting from CA2 activation could be detected in PFC (*Figure 3*). Using dual recordings from hippocampus and PFC, with implanted wire electrodes targeting prelimbic cortex (see *Figure 2—figure supplement 1*), we found that CNO treatment induced significant increases in low-gamma power in PFC during both run and rest periods (N = 4; run: F(1.177, 3.53) = 9.154, p = 0.0444; rest: (1.561, 4.6 84) =4.684, p = 0.0409, repeated-measures one-way ANOVA with Geisser-Greenhouse correction for unequal variance, *Figure 3B–C*). Theta and high-gamma powers were not affected by CNO treatment (data not shown). Control *Amigo2*-icreERT2- animals showed no significant change in PFC low-gamma power following CNO administration (N = 4; run: F(1.349, 4.047) = 1.809, p = 0.2617; *Figure 3D* and *Figure 3—figure supplement 1*). Further, we detected no significant changes in low-gamma power in *Amigo2*-icreERT2+ animals implanted with wire electrodes that missed their PFC target (N = 3; run: F(1.742, 3.483) = 0.7609, p = 0.5145; repeated-measures one-way ANOVA with Geisser-Greenhouse correction for unequal variance; *Figure 3E*, *Figure 2—figure supplement 1C*;

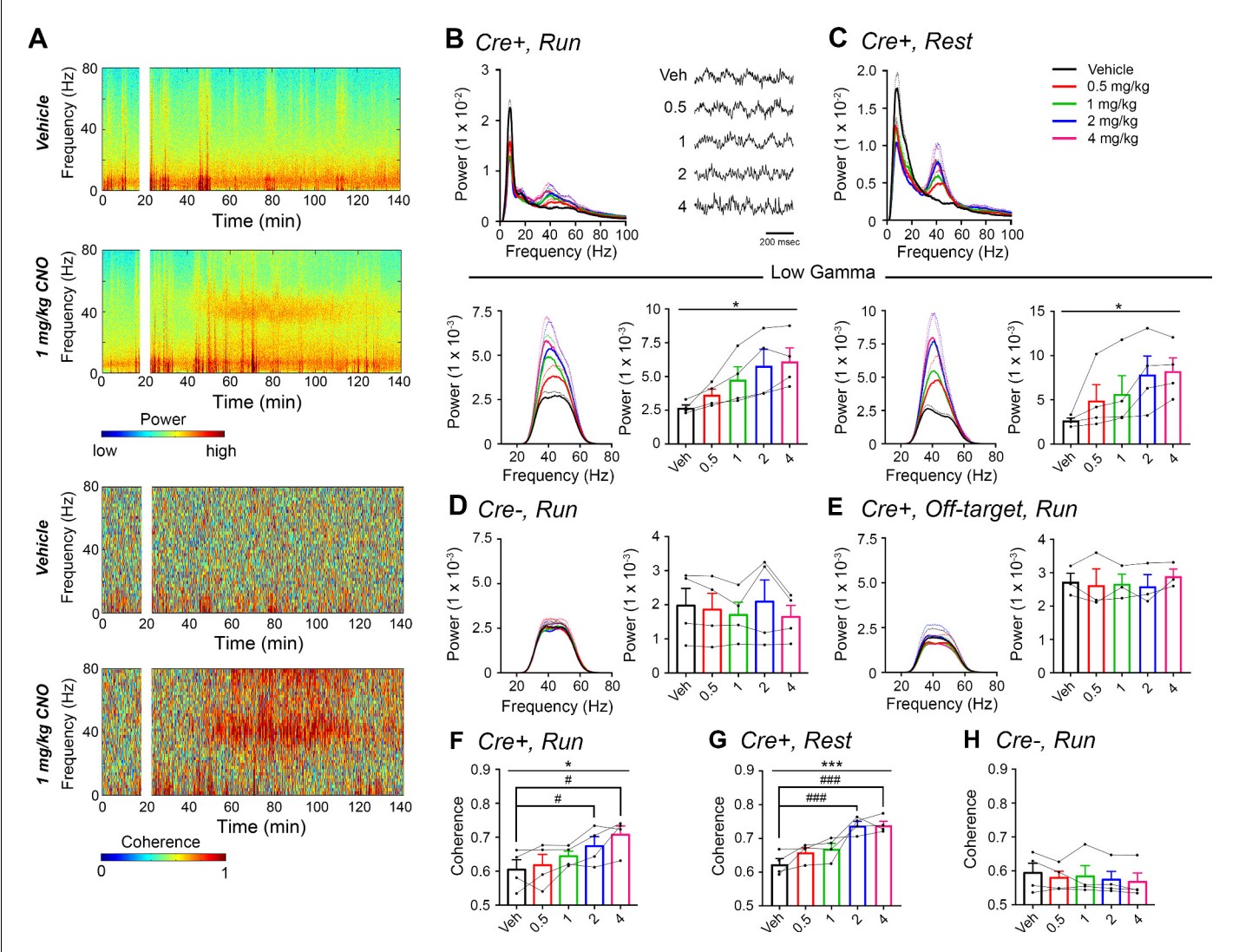

**Figure 3.** CNO treatment dose-dependently increases low-gamma power in PFC of hM3Dq-infused *Amigo2*-icreERT2+ mice. (**A**) Spectrograms of PFC LFP recordings depicting power (top two panels) and coherograms depicting coherence between PFC and hippocampal LFP recordings (bottom two panels) according to different frequencies over time. Vehicle/CNO (1 mg/kg, SQ) administration time is shown by the white bar, and the treatment is shown to the left of each spectrogram. (**B–C**) Power measures for PFC LFPs during periods of running (**B**) and resting (**C**) for *Amigo2*-icreERT2+ mice. For each of B and C: Top plots show power spectral densities of LFPs for frequencies up to 100 Hz and bottom plots show power measured from PFC LFPs filtered in the low-gamma (30 – 60 Hz) frequency range for each of run and rest periods. LFP traces in B show example LFPs during periods of running following the listed treatment. CNO treatment significantly increased low-gamma power during both running (N = 4 mice (three male, one female); $F_{(1.177, 3.53)}$ = 9.154, p = 0.0444, repeated-measures one-way ANOVA with Geisser-Greenhouse correction for unequal variance; results of Holm-Sidak post hoc tests not significant) and resting ($F_{(1.561, 4.684)}$ = 7.155), p = 0.0409, repeated-measures one-way ANOVA with Geisser-Greenhouse correction for unequal variance). (**D**) Mean low-gamma power spectra and peak low-gamma power recorded from PFC for the population of hM3Dq infused *Amigo2*-icreERT2- mice during periods of running. Low-gamma power did not significantly change in *Amigo2*-icreERT2- mice upon CNO administration (N = 4 male mice; $F_{(1.349, 4.047)}$ = 1.809, p = 0.2617; repeated-measures one-way ANOVA with Geisser-Greenhouse correction for unequal variance). (**E**) Mean low-gamma power spectra and peak low-gamma power for recordings from *Amigo2*-icreERT2+ mice infused with hM3Dq in which recording wires missed the target PFC area. CNO administration produced no significant change in peak low-gamma power from off-target recordings (N = 3 mice (one male, two female); $F_{(1.742, 3.483)}$ = 0.7609, p = 0.5145; repeated-measures one-way ANOVA with Geisser-Greenhouse correction for unequal variance). Each of the animals used for data shown in E showed increased low-gamma power in hippocampus upon CNO administration. (**F–G**) Mean low-gamma coherence between PFC and hippocampal during periods of run (**F**) and rest (**G**) for *Amigo2*-icreERT2+ mice successfully targeted to PFC. CNO treatment produced a significant increase in low-gamma coherence between hippocampus and PFC during both running (N = 4 mice; $F_{(1.595, 4.786)}$ = 8.279, p = 0.0305; repeated-measures one-way ANOVA with Geisser-Greenhouse correction for unequal variance, results of Holm-Sidak post hoc tests shown by symbols) and resting ($F_{(4, 12)}$ = 11.71, p = 0.0004; repeated-measures one-way ANOVA, results of Holm-Sidak post hoc tests shown by symbols). (**H**) *Amigo2*-icreERT2- animals showed no significant change in low-gamma coherence upon CNO

*Figure 3 continued on next page*

*Figure 3 continued*

administration (N = 4 male mice; F(4, 12) = 1.053, p = 0.4209; repeated-measures one-way ANOVA). All spectral plots show mean spectra for the population of animals with colors representing treatments. Bar graphs show mean peak gamma power (B–E) or mean gamma coherence (F–H) for the population of animals in colored bars according to treatment and data from individual animals in black dots. Dotted lines on spectral plots and error bars on bar graphs represent standard error of the mean. *p < 0.05, ***p < 0.001, one-way ANOVA; #p < 0.05, ### p < 0.001, Holm-Sidak post hoc test. See also *Figure 3—figure supplements 1–2*.

DOI: https://doi.org/10.7554/eLife.38052.014

The following figure supplements are available for figure 3:

**Figure supplement 1.** CNO did not affect low-gamma power in PFC of hM3Dq-infused Cre- *Amigo2*-icreERT2 mice.

DOI: https://doi.org/10.7554/eLife.38052.015

**Figure supplement 2.** CNO did not affect low-gamma power when PFC-targeted electrodes did not hit PFC in hM3Dq-infused Cre+ *Amigo2*-icreERT2 mice.

DOI: https://doi.org/10.7554/eLife.38052.016

*Figure 3—figure supplement 2*) despite those animals showing increased low-gamma power in hippocampus (N = 3; F(1.39, 2.781) = 81.51, p = 0.0036; repeated-measures one-way ANOVA with Geisser-Greenhouse correction for unequal variance). In addition, analysis of low-gamma power in *Amigo2*-icreERT2+, *Amigo2*-icreERT2-, and off-target implanted *Amigo2*-icreERT2+ animals showed a significant interaction between treatment and animal group (F(8, 32) = 7.25, p < 0.0001; two-way ANOVA), with significant increases in low-gamma power occurring only in *Amigo2*-icreERT2+ animals with on-target electrode placement (0.5 mg/kg: p = 0.1421, 1 mg/kg: p = 0.0002, 2 mg/kg: p < 0.0001, 4 mg/kg: p < 0.0001; Bonferroni's multiple comparisons test). These findings indicate that the increase in gamma power we detected in PFC was not due to electrical artifact or brain-wide changes in activity but rather to specific hippocampal inputs into the PFC (*Thierry et al., 2000*; *Swanson, 1981*).

Because we found increased low-gamma power upon CNO administration in both hippocampus and PFC, we analyzed LFP coherence between the two signals to measure the extent to which the two brain areas oscillated together. CNO administration produced a significant increase in low-gamma coherence between hippocampus and PFC during both run (N = 4; F(1.595, 4.786) = 8.279, p = 0.0305; repeated-measures one-way ANOVA with Geisser-Greenhouse correction for unequal variance; results of Holm-Sidak post hoc tests versus vehicle: 0.5 mg/kg: p = 0.5808, 1 mg/kg: p = 0.2079, 2 mg/kg: p = 0.0292, 4 mg/kg: p = 0.0292; *Figure 3F*), and rest (F(4, 12) = 11.71, p = 0.0004; repeated measured one-way ANOVA; results of Holm-Sidak post hoc tests versus vehicle: 0.5 mg/kg: p = 0.1189, 1 mg/kg: p = 0.1018, 2 mg/kg: p = 0.0006, 4 mg/kg: p = 0.0006; *Figure 3G*). In contrast, treatment with CNO produced no significant change in coherence between hippocampus and PFC in control *Amigo2*-icreERT2- animals (N = 4; F(4, 12) = 1.053, p = 0.4209; repeated-measures one-way ANOVA; *Figure 3H*).

## Increasing CA2 pyramidal cell activity decreases sharp-wave ripple occurrence

CA2 neuronal activity was recently reported to ramp up before the onset of sharp-wave ripples (*Oliva et al., 2016*), so we were interested in whether and how modifying CA2 neuronal activity would impact sharp-wave ripples recorded in CA1. Therefore, we measured ripple oscillations from the CA1 pyramidal cell layer of *Amigo2*-icreERT2+ and control *Amigo2*-icreERT2- mice infused with hM3Dq during periods of rest 30 – 60 min following administration of either CNO (0.5 mg/kg, SQ; *Figure 4*) or vehicle as control. We chose to use a low dose of CNO in this experiment to minimize the possibility that ripple-filtered LFPs would be contaminated by neuronal spiking in response to CNO administration independent of ripple-associated spiking. In *Amigo2*-icreERT2+ animals, CNO administration significantly decreased ripple event rate relative to that observed following vehicle administration (N = 8; t(7) = 4.574, p = 0.0026; two-tailed paired t-test; *Figure 4C*), although ripple amplitude was not significantly affected (t(7) = 0.3004, p = 0.7726; two-tailed paired t-test; *Figure 4D*). In control *Amigo2*-icreERT2- animals, CNO administration had no effect on ripple event rate or amplitude (N = 4; event rate: t(3) = 1.871, p = 0.1581; amplitude: t(3) = 0.3193, p = 0.7704; two-tailed paired t-test; *Figure 4E–F*). Further, analysis of ripple event rate in *Amigo2*-icreERT2+ and *Amigo2*-icreERT2- animals together showed a significant interaction between

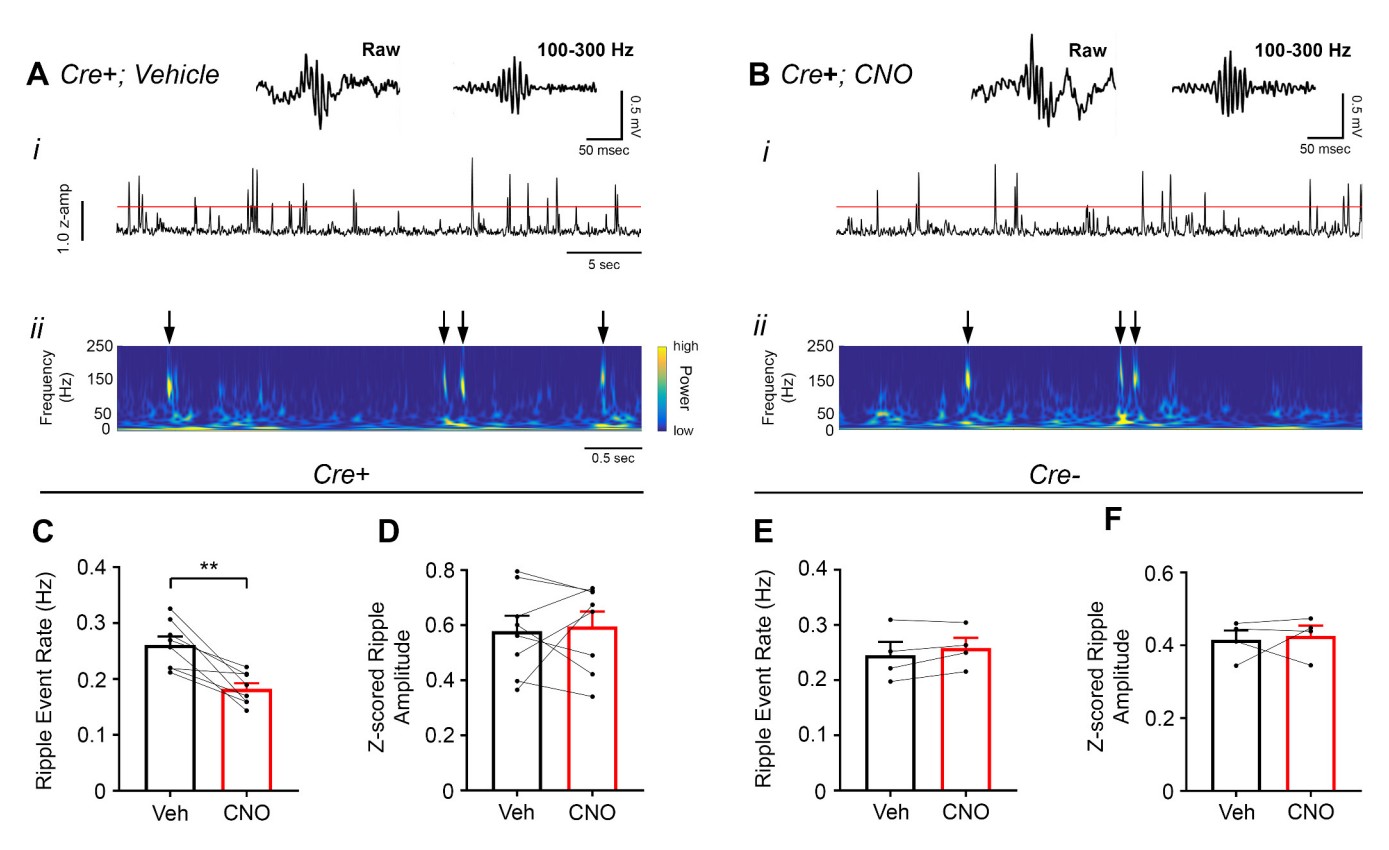

**Figure 4.** Chemoactivation of CA2 pyramidal cells with hM3Dq decreases high-frequency ripple event rate. (A–B) Envelopes of ripple-filtered CA1 LFPs (i) recorded during periods of rest following administration of vehicle (A) or CNO (B; 0.5 mg/kg, SQ) and wavelet-filtered spectrograms (ii) of the same LFPs. Cooler colors represent low power and warmer colors represent high power. Arrows denote examples of ripples shown by spectrogram. Raw and filtered LFPs showing example ripples following vehicle or CNO are shown on top. (C–D) Ripple event rate (C) but not amplitude (D) was significantly decreased in hM3Dq-expressing mice following CNO administration compared to that following vehicle administration (Ripple event rate: N = 8 mice (five male, three female); t(7) = 4.574, p = 0.0026; two-tailed paired t-test; Amplitude: t(7) = 0.3004, p = 0.7726, two-tailed paired t-test). (E–F) Ripple event rate and amplitude were not significantly changed in *Amigo2*-icreERT2- hM3Dq-infused mice (Ripple event rate: N = 4 male mice, t(3) = 1.871, p = 0.1581, two-tailed paired t-test; Amplitude: N = 4 male mice, t(3) = 0.3193, p = 0.7704, two-tailed paired t-test). **p < 0.01.

DOI: https://doi.org/10.7554/eLife.38052.017

treatment and genotype (F(1, 10) = 12.93, p = 0.0049; two-way ANOVA), with only *Amigo2*-icreERT2+ animals showing a significant decrease in ripple event rate (p = 0.0007; Bonferroni's multiple comparisons test).

## CA2 pyramidal cell inhibition decreases hippocampal and prefrontal cortical low-gamma power

Based on our finding that increasing activity of CA2 neurons in hM3Dq-expressing mice increased low-gamma power in hippocampus and PFC, we hypothesized that inhibition of CA2 pyramidal neurons with hM4Di would decrease gamma power. As a control experiment to ascertain whether hM4Di would decrease CA2 synaptic output in our system, we infused *Amigo2*-icreERT2+ mice with AAV-EF1a-DIO-hChR2(H134R)-EYFP (ChR2) and hM4Di AAVs, treated animals with tamoxifen, and then implanted the animals with fiber optic probes in CA2 and electrode bundles in the ipsilateral intermediate CA1. Optogenetic stimulation of CA2 in these awake, behaving animals evoked detectable voltage responses in CA1 that were inhibited as early as 20 min post CNO treatment (the earliest we tested). In this preparation, we detected inhibition of CA2 responses for 4 hr. By 24 hr, responses recovered to 77.20% of pre-CNO response amplitude (*Figure 5—figure supplement 1*).

To test our hypothesis that hM4Di inhibition of CA2 output would decrease hippocampal and prefrontal cortical low-gamma power, we recorded LFPs from *Amigo2*-icreERT2+ and control *Amigo2*-icreERT2- mice infused with hM4Di AAV, treated with tamoxifen and implanted with electrodes. Hippocampal LFPs were measured from the primary target of CA2 pyramidal neurons, CA1 (four mice with dorsal CA1 electrodes, four mice with intermediate CA1 electrodes, *Figure 5—figure supplement 2*), because the majority of the neuronal inhibition by hM4Di occurs at the axon terminal to reduce neurotransmitter release (*Stachniak et al., 2014*). Using identical analyses as for hM3Dq-infused animals, we compared LFPs filtered in the theta (5 – 10 Hz), low-gamma (30 – 60 Hz) and high-gamma (65 – 100 Hz) frequency ranges during periods of running and resting 30 – 60 min following administration of CNO (5 mg/kg, SQ) or vehicle. We found a significant decrease in low-gamma power during running following CNO administration compared with vehicle (t(7) = 4.408, p = 0.0031, two-tailed paired t-test, *Figure 5Aiii*, *Figure 5F*). Because recordings were made from dorsal CA1 in half of the animals and intermediate CA1 in half of the animals (*Figure 5—figure supplement 2*), we separated animals into two groups based on electrode location and compared changes in low-gamma power for the two groups (*Figure 5—figure supplement 3*). We again found a main effect of drug treatment during periods of running but no significant difference in the magnitude of low-gamma power change between recordings from dorsal CA1 and recordings from intermediate CA1 during periods of running (main effect of treatment: F(1, 6) = 17.84, p = 0.0055; main effect of location: F(1, 6) = 1.002, p = 0.3555; interaction: F(1, 6) = 0.428, p = 0.5372; *Figure 5—figure supplement 3A–B*) or periods of rest (main effect of treatment: F(1, 6) = 0.3002, p = 0.6035; main effect of location: F(1, 6) = 0.2614, p = 0.6274; interaction: F(1, 6) = 4.278, p = 0.0841; *Figure 5—figure supplement 3C–D*). To address whether difference in locomotor activity between treatments could account for the difference in low-gamma power, we measured the percent of time running and the mean running velocity following treated with either vehicle or CNO. We found no difference in either of these measures (percent of time running: main effect of genotype: F(1, 12) = 0.02054, p = 0.8884; main effect of treatment: (1, 12) = 2.988; interaction (F(1, 12) = 0.02761, p = 0.8708; running velocity: main effect of genotype: F(1, 12) = 0.2934, p = 0.5979; main effect of treatment: F(1, 12) = 2.465, p = 0.1424; interaction: (F(1, 12) = 0.02162, p = 0.8855; two-way ANOVA; *Figure 2—figure supplement 4C–D*). We also measured theta phase, low-gamma amplitude comodulation during periods of running and found that theta phase of peak gamma amplitude was not significantly affected by CNO but modulation index was significantly decreased with CNO treatment (N = 8, phase: t(7) = 1.000, p = 0.3506, two-tailed paired t-test; modulation index: *W* = -32, p = 0.0234, Wilcoxon signed-ranked test; *Figure 5—figure supplement 4A*).

Treatment with CNO did not affect theta or high-gamma power during running (theta: t(7) = 0.7786, p = 0.4617; high-gamma: t(7) = 2.029, p = 0.0821) and did not affect power in any of these frequency bands during periods of rest (theta: t(7) = 2.214, p = 0.0625; low-gamma: (t(7) = 0.4522, p = 0.6648; high-gamma: t(7) = 0.172, p = 0.8683; *Figure 5A–B*). To assess whether gamma power changes were specific to the low-gamma range, we also compared the change in gamma power for low- and high-gamma together. This analysis confirmed the selective decrease in low-gamma power during running (main effect of treatment: F(1, 7) = 18.38, p = 0.0036; main effect of gamma range: F(1,7) = 14.57, p = 0.0066; interaction: F(1, 7) = 10.64, p = 0.0138, two-way ANOVA; low-gamma, p = 0.0008; high-gamma, p = 0.2703, Bonferroni multiple comparison tests; *Figure 5—figure supplement 5A*) and also confirmed the lack of decrease in gamma power during periods of rest (main effect of treatment: F(1, 7) = 0.1067, p = 0.7535; main effect of gamma range: F(1,7) = 41.65, p = 0.0003; interaction: F(1, 7) = 41.65, p = 0.6433, two-way ANOVA; low- and high-gamma: p > 0.9999; *Figure 5—figure supplement 5B*).

CA2 pyramidal neurons have been shown to possess axons with large rostral to caudal trajectories, primarily targeting CA1 (*Tamamaki et al., 1988*). Consistent with a projection toward caudal CA1, we observed fluorescence from hM4Di-mCherry+ axon fibers in caudal intermediate CA1, with most of the fluorescently-labeled CA2 axons targeting *stratum oriens* in CA1 (*Kohara et al., 2014*) (*Figure 5C*). Ventral CA1 neurons, in turn, project to PFC (*Swanson, 1981*; *Jay and Witter, 1991*; *Jay et al., 1989*). Therefore, we asked whether inhibition of CA2 pyramidal neurons would impact low-gamma power recorded in PFC. A subset of *Amigo2*-icreERT2+ and control *Amigo2*-icreERT2- mice with electrodes in CA1 were also implanted with electrodes in PFC and treated with CNO (5 mg/kg, SQ) or vehicle as control. In *Amigo2*-icreERT2+ mice, we observed a small but consistent decrease in PFC low-gamma power during running following CNO administration compared with

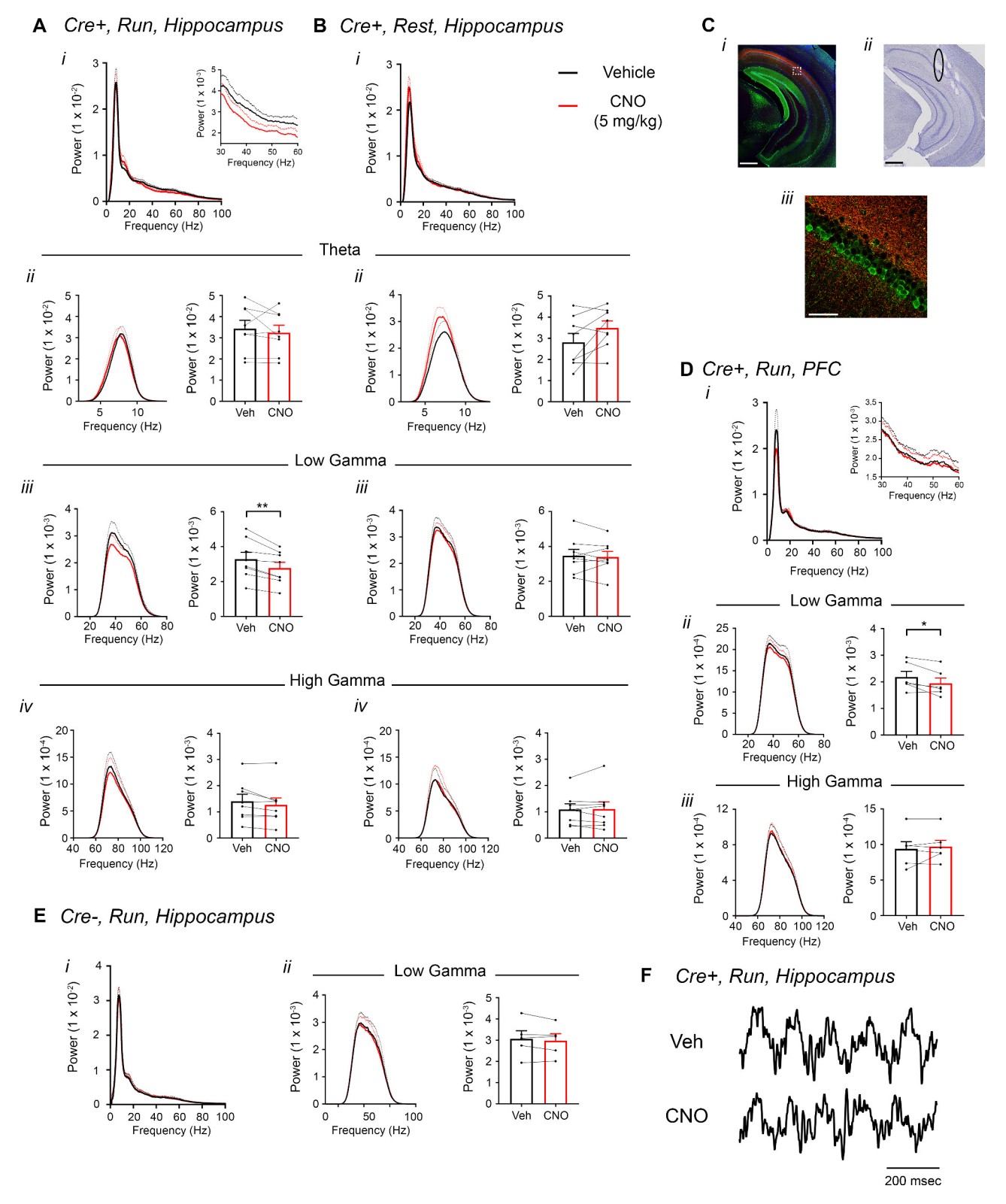

**Figure 5.** Inhibition of CA2 pyramidal cells with hM4Di decreases hippocampal and PFC low-gamma power. (A–B) Hippocampal LFP power measures from *Amigo2*-icreERT2+ mice infused with hM4Di AAV and treated with vehicle or CNO (5 mg/kg, SQ; LFP samples 30 – 60 min following treatment) during periods of running (A) and resting (B). For each of A and B: (*i*) Power spectral density plots from raw LFPs for frequencies up to 100 Hz. Inset plot in A*i* is expanded from A*i*. (*ii-iv*) Power spectral density plots and peak power measured in the theta (5 – 10 Hz; *ii*), low-gamma (30 – 60 Hz; *iii*) or high-
*Figure 5 continued on next page*

*Figure 5 continued*

gamma (65 – 100 Hz; *iv*) frequency ranges. In A*ii-iv* and B*ii-iv*, plots on the left show power spectral density for the listed frequency bands, and plots on the right show mean peak power for the population of animals in colored bars according to treatment and dots representing data from individual animals. (A*iii*) CNO administration produced a significant decrease in hippocampal low-gamma power during running (N = 8 mice (four female, four male); t(7) = 4.408, p = 0.0031, two-tailed paired t-test). CNO treatment did not affect theta power during running (t(7) = 0.7786, p = 0.4617; A*ii*), high-gamma power during running (t(7) = 2.029, p = 0.0821; A*iv*), theta power during rest (t(7) = 2.214, p = 0.0625; B*ii*), low-gamma power during rest (t (7) = 0.4522, p = 0.6648; B*iii*) or high-gamma power during rest (t(7) = 0.172, p = 0.8683; B*iv*). (C) Expression of mCherry-tagged hM4Di in intermediate CA1 and electrode tracks at a similar position. (C*i*) Expression of mCherry-tagged hM4Di (red) and calbindin (green, a marker for superficial CA1 neurons (*Kohara et al., 2014*) in an intermediate hippocampal section. The white box shows the area that is expanded in C*iii*. Axons expressing hM4Di target intermediate CA1, with preferential targeting toward *stratum oriens*. (C*ii*) Electrode tracks of intermediate CA1 recording wires (black ellipse surrounds one of the tracks). (D) PFC LFP power measures from same mice used in A-B. (i) Power spectral density plots from raw LFPs for frequencies up to 100 Hz. Inset plot is expanded from the adjacent plot. (*ii-iii*) Power spectral density plot and peak power measured from low-gamma (*ii*) and high-gamma (*iii*) filtered LFPs. CNO administration produced a significant decrease in PFC low-gamma power during running (N = 6 mice (three female, three male); t(5) = 2.948, p = 0.0320, two-tailed paired t-test) but did not affect PFC high-gamma power (t(5) = 0.738, p = 0.4937. (E) Power spectral density plot for the population of *Amigo2*-icreERT2- mice infused with hM3Di, treated with tamoxifen and challenged with CNO. Plots show spectral density of frequencies below 100 Hz (i), low-gamma-filtered LFP spectral power and peak low-gamma power for the population of animals (ii). CNO administration did not significantly affect low-gamma power in *Amigo2*-icreERT2- mice during running (N = 5 male mice; t(4) = 1.079, p = 0.3413; two-tailed paired t-test). (F) Example LFP traces from periods of running following vehicle or CNO treatment. Both recordings are taken from the same recording site in the same animal. *p < 0.05, **p < 0.01. Scale bars = 500 um (C*i, ii*) and 75 um (C*iii*). See also *Figure 5—figure supplements 1–5*.
DOI: https://doi.org/10.7554/eLife.38052.018

The following figure supplements are available for figure 5:

**Figure supplement 1.** Optogenetic activation of CA2 neurons evokes responses in intermediate CA1, which are blocked by hM4Di.
DOI: https://doi.org/10.7554/eLife.38052.019

**Figure supplement 2.** Electrode localization for Cre+ and Cre- *Amigo2*-icreERT2 mice infused with hM4Di AAV and implanted with electrodes into hippocampus and PFC.
DOI: https://doi.org/10.7554/eLife.38052.020

**Figure supplement 3.** The magnitude of hM4Di-mediated change in low-gamma power did not vary significantly as a function of dorsal or intermediate recording location for periods of running (**A–B**) of rest (**C–D**).
DOI: https://doi.org/10.7554/eLife.38052.021

**Figure supplement 4.** Comodulation of low-gamma amplitude and theta phase in hM3Di mice with vehicle or CNO treatment.
DOI: https://doi.org/10.7554/eLife.38052.022

**Figure supplement 5.** Change in gamma power by hM4Di is selective for low gamma and does not significantly affect high-gamma during running.
DOI: https://doi.org/10.7554/eLife.38052.023

vehicle (N = 6; t(5) = 2.948, p = 0.0320; two-tailed paired t-test; *Figure 5D*), suggesting that CA2 activity modulates PFC low-gamma oscillations, likely via intermediate CA1. We found no change in PFC theta or high gamma power (theta: t(5) = 1.341, p = 0.2375; high gamma: t(5) = 0.738, p = 0.4937, paired t-test; *Figure 5Diii*). Control *Amigo2*-icreERT2- animals showed no significant change in hippocampal or PFC low-gamma power in response to CNO treatment compared with vehicle (N = 5; hippocampus: t(4) = 1.079, p = 0.3413, two-tailed paired t-test; PFC: t(4) = 0.4293, p = 0.6898, two-tailed paired t-test; *Figure 5E*, *Figure 5—figure supplement 5C–D*) and no change in hippocampal theta phase, low-gamma amplitude comodulation (N = 5, phase: t(4) = 1.206, p = 0.2943, two-tailed paired t-test; modulation index: t(4) = 0.206, p = 0.8469, two-tailed paired t-test; *Figure 5—figure supplement 4B*). Further, analysis of hippocampal low-gamma power in *Amigo2*-iCreERT2+ animals and *Amigo2*-iCreERT2- animals together showed a significant interaction between treatment and genotype (F(1, 11) = 6.788, p = 0.0245, two-way ANOVA), with only *Amigo2*-iCreERT2+ animals showing a significant change in low-gamma power with CNO treatment (p = 0.0007, Bonferonni's multiple comparisons test).

## CA2 pyramidal cell inhibition increases hippocampal ripple oscillations

To assess the influence of inhibiting CA2 output on ripple oscillations, we measured ripples from the CA1 pyramidal cell layer in *Amigo2*-icreERT2+ and control *Amigo2*-icreERT2- mice during periods of rest, 30 – 60 min following administration of either CNO (5 mg/kg, SQ) or vehicle control (*Figure 6*). As predicted based on our findings in hM3Dq-infused animals, CNO administration significantly increased ripple event rate in hM4Di-infused animals (N = 6; t(5) = 3.809, p = 0.0063; one-tailed paired t-test; *Figure 6C*). CNO administration also increased ripple amplitude in hM4Di animals (t

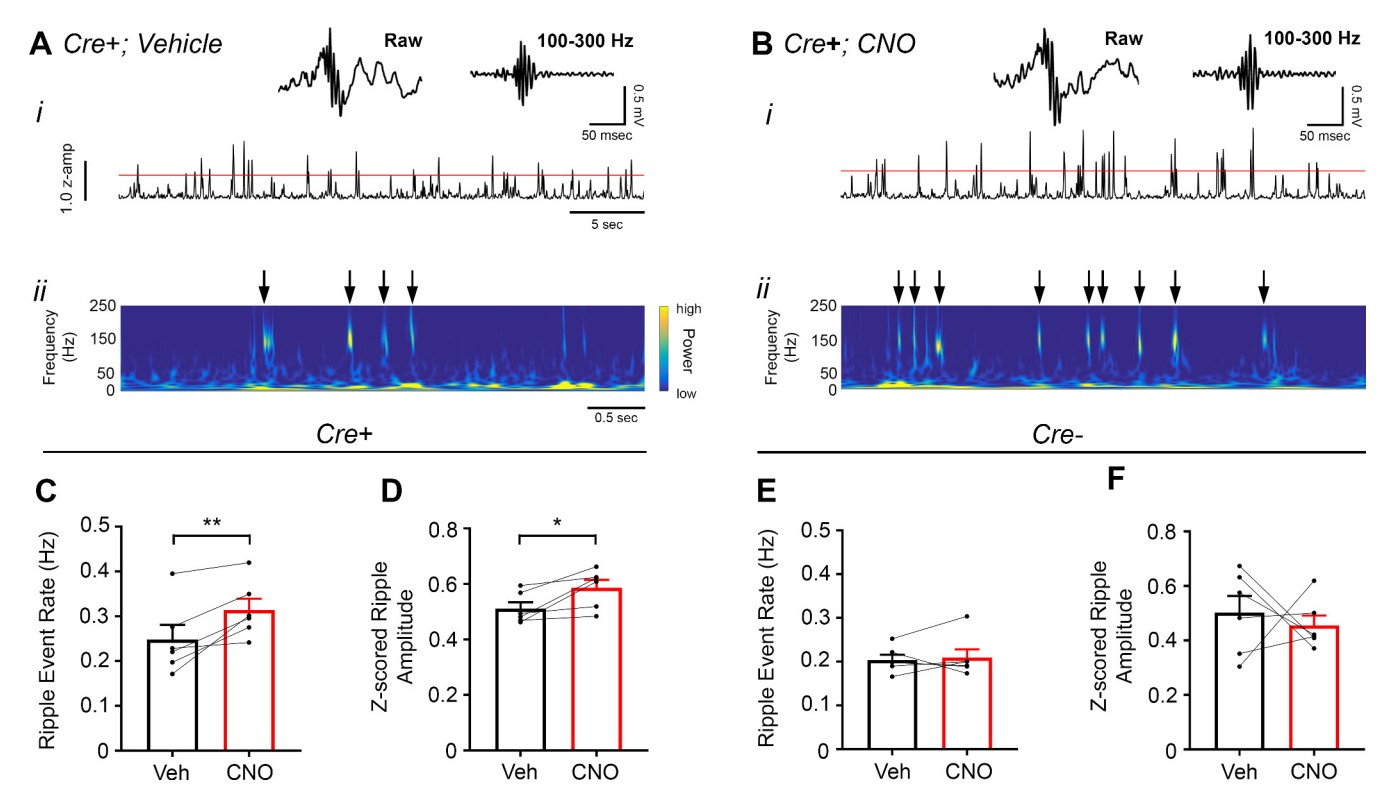

**Figure 6.** Inhibition of CA2 pyramidal cells with hM4Di increases high-frequency ripple event rate and amplitude. (A–B) Envelopes of ripple-filtered CA1 LFPs (*i*) recorded during periods of rest following administration of vehicle (A) or CNO (B; 5 mg/kg, SQ) and wavelet-filtered spectrograms (*ii*) of the same LFPs. Cooler colors represent low power and warmer colors represent high power. Arrows denote examples of ripples shown by spectrogram. Raw and filtered LFPs of example ripples following vehicle or CNO are shown on top. (C–D) Ripple event rate (C) and amplitude (D) were significantly increased in hM4Di-expressing mice following CNO treatment compared to that following vehicle treatment (Ripple event rate: N = 6 mice (three male, three female); t(5) 3.809, p = 0.0063; two-tailed paired t-test; Amplitude: N = 6; t(5) = 3.069, p = 0.0278, two-tailed paired t-test). (E–F) Ripple event rate and amplitude were not significantly changed in *Amigo2*-icreERT2- hM4Di-infused mice (Ripple event rate: N = 6 male mice; W = 5, p = 0.6875, Wilcoxon signed-ranked test; Amplitude: t(5) = 0.5165, p = 0.6275, two-tailed paired t-test). *p < 0.05, **p < 0.01.cortical network. Further, these findings support a negative regulatory role of CA2 in hippocampal sharp-wave ripples.
DOI: https://doi.org/10.7554/eLife.38052.024

(p = 0.0278; two-tailed paired t-test; *Figure 6D*). By contrast, in control *Amigo2*-icreERT2- mice, CNO administration did not significantly change ripple event rate or amplitude (N = 6; event rate: W = 5, p = 0.6875, Wilcoxon signed-ranked test; amplitude: t(5) = 0.5165, p = 0.6275, two-tailed paired t-test; *Figure 6E–F*). In addition, analysis of ripple event rate in *Amigo2*-iCreERT2+ animals and *Amigo2*-iCreERT2- animals together showed a significant interaction between treatment and genotype (F(1, 10) = 7.215, p = 0.0229, two-way ANOVA), with only *Amigo2*-iCreERT2+ animals showing a significant change in ripple event rate with CNO treatment (p = 0.0041, Bonferonni's multiple comparisons test). These data, together with our hM3Dq ripple findings, indicate that hippocampal ripple occurrence is negatively modulated by activity of CA2 pyramidal neurons.

## Discussion

In this study, we used excitatory and inhibitory DREADDs to reversibly modify activity of CA2 pyramidal cells and examined the effect on hippocampal and prefrontal cortical oscillations and behavior. We found that increasing activity of CA2 pyramidal cells increased hippocampal and prefrontal cortical low-gamma power and decreased hippocampal sharp-wave ripple events. Conversely, we found that inhibiting CA2 pyramidal cell output decreased hippocampal and prefrontal cortical low-gamma power and increased hippocampal sharp-wave ripples. These findings demonstrate a role for

hippocampal area CA2 in low-gamma oscillation generation across the distributed hippocampal-prefrontal cortical network. Further, these findings support a negative regulatory role of CA2 in hippocampal sharp-wave ripples.

CA2 activation produced robust, dose-dependent increases in hippocampal low-gamma power, and inhibition of CA2 neurons decreased low-gamma power. Gamma oscillations in hippocampus and cortex reflect synchronous inhibitory postsynaptic potentials (IPSPs) from a network of interconnected perisomatically targeted basket cells, with excitatory drive onto these interneurons arising from pyramidal cells (*Fisahn et al., 1998*; *Mercer et al., 2012*; *Valero et al., 2015*; *Lee et al., 2014*). The frequency of the gamma oscillations is thought to be controlled by the decay kinetics of the IPSP such that slower decay yields a lower gamma frequency (*Fisahn et al., 1998*).

Low-gamma oscillations in CA1 were previously reported to arise from neuronal activity in CA3 (*Colgin et al., 2009*; *Schomburg et al., 2014*; *Alexander et al., 2009*; *Bragin et al., 1995*). However, permanent silencing of CA3 output with tetanus toxin light chain produced only a 30% reduction in low-gamma power recorded in CA1 (*Middleton and McHugh, 2016*), thereby challenging the notion that CA3 is the only origin of hippocampal low-gamma oscillations. Here, we report that CA2 activation increases low-gamma power and that acute CA2 silencing reduces low-gamma power recorded in CA1 by approximately 20%. CA2 pyramidal cells target local inhibitory basket cells in CA2 (*Mercer et al., 2012*), providing a mechanism for local gamma generation within CA2, as well as similar inhibitory neurons in CA1, permitting feed-forward excitation to drive IPSPs and gamma oscillations in CA1 (*Valero et al., 2015*). hM4Di was previously reported to substantially, but not entirely, inhibit synaptic neurotransmitter release (*Stachniak et al., 2014*), although we found that hM4Di produced near complete silencing of CA2 output (*Figure 5—figure supplement 1*). Complete silencing of CA2 with tetanus toxin light chain may reduce low-gamma power by greater than the 20% we observed here, but more likely, silencing of both CA2 and CA3 may be needed to abolish low-gamma power. Based on these findings, CA2 and CA3 together likely provide the excitatory drive required to generate low-gamma oscillations in CA1. Given the dynamic nature of the brain, we propose that when CA2 is inhibited, CA3, or another source, is capable of compensating, and vice versa. Further, gamma activity arising from CA2 and CA3 may engage distinct circuits involving the deep and superficial CA1 pyramidal neurons, respectively (*Kohara et al., 2014*; *Lee et al., 2014*). As such, low-gamma oscillations arising from the two areas may subserve distinct cognitive functions based on the output of these two populations. In addition, the hippocampus is known to function in spatial navigation, and navigation strategies may be either cue-based, relying on sensory cued or landmarks to navigate (*Gothard et al., 1996*), or sequence-based, relying on sequential steps or turns to navigate (*Rondi-Reig et al., 2006*). Low-gamma oscillations, arising from CA2 and CA3 appear to be related to sequence-based navigation, consistent with a role in memory recall (*Colgin et al., 2009*; *Tort et al., 2009*; *Cabral et al., 2014*). Therefore, the low-gamma oscillations generated by CA2 neurons may contribute to sequence-based spatial navigation.

Modification of CA2 neuronal activity also affected the occurrence of sharp-wave ripple oscillations. Specifically, increasing CA2 activity with hM3Dq decreased the occurrence of ripples, whereas decreasing CA2 output with hM4Di increased ripple occurrence as well as amplitude. The mechanism underlying these findings likely includes the robust inhibition that CA2 presents over CA3 neurons (*Boehringer et al., 2017*; *Kohara et al., 2014*). Accordingly, CA2 pyramidal cells contact local parvalbumin-expressing basket cells, which project to CA3 (*Mercer et al., 2012*), and CA2 pyramidal cell firing is reported to discharge CA3 interneurons (*Oliva et al., 2016*). As a potential secondary mechanism underlying the observed inverse relationship between CA2 neuronal activity and occurrence of ripples, CA2 neurons preferentially target the deep layer of pyramidal cells in CA1 (*Kohara et al., 2014*). During recordings of ripples from these deep CA1 pyramidal cells, dominant hyperpolarizations are observed, which contrasts with dominant depolarizations during ripples seen in superficial CA1 pyramidal cells (*Valero et al., 2015*; see also *Lee et al., 2014*). Further, stimulation of CA2 neurons produces robust feed-forward inhibitory responses onto CA1 neurons (*Boehringer et al., 2017*; *Valero et al., 2015*). Therefore, silencing of CA2 neurons may remove feed-forward inhibition and produce a net excitatory response in CA1. Together, these two mechanisms may explain the significant gating influence that CA2 neurons have over hippocampal excitability and, consequently, sharp-wave ripples in CA1.

Consistent with this finding, mice in which CA2 synaptic output was fully and permanently blocked via tetanus toxin light chain showed normal ripples during immobility and also anomalous

epileptiform discharges that arose from CA3 (*Boehringer et al., 2017*). Our findings of increased ripples in CA1 upon acute silencing of CA2 output are consistent with these findings in that in both studies, CA2 silencing increases CA3 to CA1 output during immobility. Echoing the statement by *Boehringer et al. (2017)*, our data do not agree with the suggestion by *Oliva et al. (2016)* that CA2 neuronal activity triggers the occurrence of ripples. Rather, CA2 activity may play a role in sculpting the CA3 network activity and gate output to CA1. Consistent with a gating, or permissive, role of CA2 toward the occurrence of ripples, revealed that CA2 is the only hippocampal subregion to have a substantial population of neurons that cease firing during CA1 ripples (*Kay et al., 2016*). Similarly, demonstrated an inverse correlation between occurrence of ripples in CA2 and CA1. During periods of low occurrence of ripples in CA2, ripple occurrence was high in CA1, and vice versa (*Oliva et al., 2016*). The inverse correlations described by these two findings suggest a negative regulatory role of CA2 activity on CA1 ripples, which is consistent with our findings.

The results of our study present further similarities between CA2 functions and impairments seen in schizophrenia. Gamma oscillations are impaired in patients with schizophrenia (*Woo et al., 2010*; *Pinkham et al., 2008*). In addition, parvalbumin-expressing interneurons, which contribute to the generation of gamma oscillations, are notably reduced in hippocampal area CA2 and PFC in post-mortem tissue from patients with this disorder (*Benes et al., 1998*; *Beasley and Reynolds, 1997*). Findings from the *Df16A±* mouse model of schizophrenia demonstrate impaired social behavior, decreased number of parvalbumin-expressing interneurons in CA2, decreased activity of CA2 pyramidal neurons (*Piskorowski et al., 2016*), and decreased synchrony between hippocampus and PFC (*Sigurdsson et al., 2010*). Additionally, the forebrain-selective calcineurin knock-out model of schizophrenia was reported to have increased CA1 ripple events during periods of resting wake (*Suh et al., 2013*). We report that CA2 neuronal activity contributes to low-gamma oscillations in PFC, which could arise from CA2 drive of CA1 neurons and, in turn, CA1 drive of excitatory and inhibitory PFC neurons, thereby contributing to low-gamma oscillation generation (*Kim and Cho, 2017*; *Marek, 2018*). We also found that CA2 activity contributes to gamma coherence between hippocampus and PFC and hippocampal ripple oscillations, suggesting that CA2 may play a role in the pathophysiology of schizophrenia.

Here, we have provided evidence that CA2 neuronal activity bidirectionally controls hippocampal and prefrontal cortical low-gamma oscillations as well as hippocampal sharp-wave ripple oscillations. These findings demonstrate a role for CA2 in the extended hippocampal-prefrontal cortical network and further support the idea that CA2 is an integral node in the hippocampal network that may be dysregulated in schizophrenia.

## Materials and methods

**Key resources table**

| Reagent type (species) or resource | Designation | Source or reference | Identifiers | Additional Information |
|---|---|---|---|---|
| Genetic reagent (*Mus musculus*) | *Amigo2*-icreERT2 | NIEHS | *B6(SJL)-Tg (Amigo2-icre/ERT2)1Ehs)* | |
| Genetic reagent (*M. musculus*) | *B6(SJL)-Tg (Amigo2-icre/ ERT2)2Ehs)* | NIEHS | *B6(SJL)-Tg (Amigo2-icre/ERT2)2Ehs)* | |
| Genetic reagent (*M. musculus*) | *ROSA*-tdTomato | Jax | B6.129S6-Gt(ROSA) 26Sor<tm9 (CAG-td Tomato) Hze>/J | RRID: Jax 007909 |
| Genetic reagent (*M. musculus*) | *GAD*-eGFP | Riken | ICR.Cg-Gad1<tm1.1Tama> | RRID: RBRC03674 |
| AAV | hM3Dq AAV | University of North Carolina-Chapel Hill Viral Vector Core | AAV-hSyn-DIO-hM3D(Gq)-mCherry | Serotype AAV5 |

*Continued on next page*

Continued

| Reagent type (species) or resource | Designation | Source or reference | Identifiers | Additional Information |
|---|---|---|---|---|
| AAV | hM4Di AAV | University of North Carolina-Chapel Hill Viral Vector Core | AAV-hSyn-DIO-hM4D(Gi)-mCherry | Serotype AAV5 |
| AAV | ChR2 AAV | University of North Carolina-Chapel Hill Viral Vector Core | AAV-EF1a-DIO-hChR2(H134R)-EYFP | Serotype AAV5 |
| Antibody | rabbit anti-PCP4 | SCBT | Cat # sc-74186 | IHC, 1:500 |
| Antibody | rabbit anti-CaMKII alpha | Abcam | Cat # ab131468 | IHC, 1:250 |
| Antibody | rat anti-mCherry | Life Technologies | Cat# M11217 | IHC, 1:500- 1:1000 |
| Antibody | mouse anti-calbindin | Swant | Cat# D-28k | IHC, 1:500 |

## Animals

Experiments were carried out in adult male and female mice (8–12 weeks at the start of experiments). Mice were housed under a 12:12 light/dark cycle with access to food and water *ad libitum*. Mice were naive to any treatment, procedure or testing at the time of beginning the experiments described here. Mice were group-housed until the time of electrode implantation for those mice undergoing electrode implantation surgery, at which point they were singly housed. All procedures were approved by the NIEHS Animal Care and Use Committee and were in accordance with the National Institutes of Health guidelines for care and use of animals. The mouse line used in this study will be made available to investigators upon request from the corresponding author .

## Generation of transgenic *Amigo2*-icreERT2

The BAC clone RP23-288P18 was used to generate these mice. To recombine the cDNA encoding an icreERT2 fusion protein (*Hainer et al., 2015*) into the BAC, we constructed a targeting vector from which we derived a targeting fragment for recombineering. The targeting fragment consisted of a 243 bp homology region (A-Box) immediately upstream of the ATG in the *Amigo2* gene. The icreERT2 cassette was fused to the A-Box replacing the *Amigo2* ATG with the icre ATG preceded with a perfect KOZAK sequence. At the 3' end of the icreERT2 cassette a synthetic bovine growth hormone (BGH) polyadenylation signal was added after the STOP codon. For selection of recombined BACs, a flipase-site flanked neomycin resistance gene was incorporated into the targeting fragment following the icreERT2 cassette. Finally, the 3' end of the targeting fragment contained a 263 bp homology region (B-Box) starting downstream of the *Amigo2* ATG. Recombineering was performed according to a previously described protocol (*Lee et al., 2001*). In brief, the targeting fragment was electroporated into induced EL250 bacteria harboring the *Amigo2* BAC. Recombined colonies were selected on Chloramphenicol/Kanamycin plates and screened by colony PCR. The neo gene was removed from the recombined BAC by arabinose driven flipase expression.

Recombined BACs without the neo marker were linearized by restriction enzyme digestion, gel purified and electro-eluted from the gel slice. After filter dialysis with a Millipore VSWP02500 filter, the BAC fragment concentration was adjusted to 1 ng/µl and microinjected into pronuclei of B6SJLF1 mouse oocytes (Taconic, North America). Six independent founder mice resulted, which were bred to *ROSA*-tdTomato indicator mice. Resulting offspring that genotyped positive for both Cre and tdTomato were treated with tamoxifen (Sigma, 100 mg/kg daily administration, IP, 7 days of treatment). At least 1 week following the final treatment with tamoxifen, mice were perfused with 4% paraformaldehyde and brains were sectioned and examined for tdTomato expression. Two lines showed adult expression of icreERT2 in CA2. One of these lines (*B6(SJL)-Tg(Amigo2-icre/ERT2)2Ehs*) showed expression of Cre in CA2 as well as sparse expression in dentate gyrus, fasciola cinerea and hypothalamus. Due to expression in the dentate gyrus, this line was not used in this study. A second line (*B6(SJL)-Tg(Amigo2-icre/ERT2)1Ehs*, referred to as *Amigo2*-icreERT2 in the text) showed selective expression in CA2 within hippocampus as well as expression in fasciola cinerea and hypothalamus, among other locations (*Figure 1A–E*) and was used for all electrophysiology and anatomy studies here. *Amigo2*-icreERT2 mice used in this study were backcrossed to C57Bl/6 seven generations.

Genotyping of *Amigo2*-icreERT2 BAC transgenic mice was done using the following primers: BGH-F (forward primer) 5'-CTT CTG AGG CGG AAA GAA CC-3' and dAmigo4 (reverse primer) 5'-AACTGCCCGTGGAGATGCTGG-3'. PCR protocol is 30 cycles of 94℃ 30 s., 60℃ 30 s., 72℃ 30 s. PCR product is 600 bp.

### Animal numbers

For all experiments presented, 39 *Amigo2*-icreERT2 mice (eight for histology, 29 for electrophysiology, two for optogenetics with electrophysiology), 13 *Amigo2*-icreERT2; *ROSA*-tdTomato mice (all for histology), 3 *Amigo2*-icreERT2; *GAD*-eGFP; *ROSA*-tdTomato mice (all for histology) and 4 *Amigo2*-icreERT2; *GAD*-eGFP mice (all for histology) were used. No statistical tests were used to determine sample sizes *a priori*, but sample sizes for histological and electrophysiological studies were similar to those used in the field (*Oliva et al., 2016*; *Boehringer et al., 2017*; *Middleton and McHugh, 2016*). For electrophysiology studies, *Amigo2*-icreERT2+ and *Amigo2*-icreERT2- animals were randomly selected from litters. For randomization, animals were housed with same-sex littermates following weaning but before genotyping. Genotype information was unknown at the time of randomly selecting a mouse from the cage for AAV infusion.

### Virus infusion and tamoxifen treatment

Viruses were obtained from the viral vector core at the University of North Carolina-Chapel Hill. Mice were infused with AAV-hSyn-DIO-hM3D(Gq)-mCherry (Serotype 5; hM3Dq AAV), AAV-hSyn-DIO-hM4D(Gi)-mCherry (Serotype 5; hM4Di AAV) or equal parts AAV-EF1a-DIO-hChR2(H134R)-EYFP (Serotype 5; ChR2 AAV) and hM4Di mixed in a centrifuge tube. For virus-infusion surgery, mice were anesthetized with ketamine (100 mg/kg, IP) and xylazine (7 mg/kg, IP), then placed in a stereotaxic apparatus. An incision was made in the scalp, a hole was drilled over each target region for AAV infusion, and a 27-ga cannula connected to a Hamilton syringe by a length of tube was lowered into hippocampus (in mm: −2.3 AP, ±2.5 ML, −1.9 mm DV from bregma). *Amigo2*-icreERT2 mice were infused unilaterally on the left side for hM3Dq AAV infusion, bilaterally for hM4Di AAV, or unilaterally on the left side for ChR2/hM4Di infusion. For each infusion, 0.5 µl was infused at a rate of 0.1 µl/min. Following infusion, the cannula was left in place for an additional 10 min before removing. The scalp was then sutured and the animals administered buprenorphine (0.1 mg/kg, SQ) for pain and returned to their cage. Two weeks following AAV infusion surgery, *Amigo2*-icreERT2 mice began daily tamoxifen treatments (100 mg/kg tamoxifen dissolved in warmed corn oil, IP) for a total of 7 days. At least 1 week following the last dose of tamoxifen, animals were euthanized and perfused with 4% paraformaldehyde for anatomical studies, or underwent electrode (and fiber optic probe for ChR2/hM4Di mice) implantation surgery.

### Electrode implantation

At least 1 week after the last tamoxifen treatment, mice for *in vivo* electrophysiology were implanted with electrode arrays. Mice were anesthetized with ketamine (100 mg/kg, IP) and xylazine (7 mg/kg, IP), then placed in a stereotaxic apparatus. An incision was made in the scalp, and the skull was cleaned and dried. One ground screw (positioned approximately 4 mm posterior and 2 mm lateral to Bregma over the right hemisphere) and four anchors were secured to the skull and electrode arrays were then lowered into drilled holes over the target brain regions. Electrode wires were connected to a printed circuit board (San Francisco Circuits, San Mateo, CA), which was connected to a miniature connector (Omnetics Connector Corporation, Minneapolis, MN). For all but one mouse that was implanted with tetrodes, electrodes consisted of stainless steel wire (44 µm) with polyimide coating (Sandvik Group, Stockholm, Sweden). Wires were bundled into groups of 8 and lowered to target regions. In 11 *Amigo2*-icreERT2 mice infused with hM3Dq AAV (7 Cre+, 4 Cre-) and 6 *Amigo2*-icreERT2 mice infused to hM4Di AAV (2 Cre+, 4 Cre-), electrode arrays were implanted into the left dorsal hippocampus, targeting CA2/proximal CA1 (in mm: −2.06 AP, −2.5 ML, −1.9 DV from bregma), the right dorsal hippocampus targeting CA1 (−1.94 AP, +1.5 ML, −1.5 DV from bregma), and the left PFC (+1.78 AP, −0.25 ML, −2.35 DV from bregma). For the sake of consistency between animals, only recordings from the left hippocampus were included in hippocampal analyses. In six *Amigo2*-icreERT2 mice infused with hM4Di AAV (4 Cre+, 2 Cre-), electrodes were lowered into left dorsal hippocampus targeting CA2/proximal CA1 (−2.06 AP, −2.5 ML, −1.9 DV

from bregma), left PFC (+1.78 AP, −0.25 ML, −2.35 DV from bregma) and left intermediate hippo-campus targeting CA1 (−2.92 AP, 2.75 ML, 2.1 DV from bregma). In three *Amigo2*-icreERT2 mice infused with hM3Dq AAV (3 Cre+), electrodes were implanted in left dorsal hippocampus targeting CA2/proximal CA1 only (−2.06 AP, −2.5 ML, −1.9 DV from bregma). In two *Amigo2*-icreERT2 mice infused with hM4Di AAV (2 Cre+), electrodes were implanted in left hippocampus targeting CA1 only (−1.94 AP,+1.5 ML, −1.25 DV from bregma). In one *Amigo2*-icreERT2+ infused with hM3Dq AAV, a bundle of eight tetrodes was lowered into the left hippocampus targeting CA2/proximal CA1 (−2.06 AP, −2.5 ML, −1.9 DV from bregma) for monitoring changes in single unit firing rate upon CNO administration. Tetrodes were constructed from 12.7 μm polyimide-coated nickel chromium (Rediohm-800) wire (Sandvic Materials Technology, Sandviken, Sweden), which were connected to a printed circuit board (San Francisco Circuits, San Francisco, CA) and miniature connector (Omnetics Connector Corporation, Minneapolis, MN). On the day of surgery, electrode tips were cut to the appropriate length and plated with gold to reduce electrode impedance to between 150 and 300 kΩ at 1 kHz by passing current through the wires while the tips were immersed in gold solution (Neuralynx, Bozeman, MT). In two *Amigo2*-icreERT2+ mice infused with ChR2/hM4Di AAV, a fiber optic probe was implanted into left CA2 (−1.95 AP, −2.25 ML, −1.65 DV from bregma) and a wire bundle was implanted into left intermediate CA1 (−3.08 AP, −2.75 ML, −2.0 DV from bregma).

## Histology

Animals used for histology were euthanized with Fatal Plus (sodium pentobarbital, 50 mg/mL; >100 mg/kg) and underwent transcardial perfusion with 4% paraformaldehyde. Brains were then cryoprotected in 30% sucrose PBS for at least 72 hr and sectioned with a cryostat or vibratome at 40 μm.

For immunohistochemistry, brain sections were rinsed in PBS, boiled in deionized water for 3 min, and blocked for at least 1 hr in 3–5% normal goat serum/0.01% Tween 20 PBS. Sections were incubated in the following primary antibodies, which have previously been validated in mouse brain (*Hitti and Siegelbaum, 2014*; *Kohara et al., 2014*: rabbit anti-PCP4 (SCBT, sc-74186, 1:500), rabbit anti-CaMKII alpha (Abcam, ab131468, 1:250), rat anti-mCherry (Life Technologies, M11217, 1:500- 1:1000), mouse anti-calbindin (Swant, D-28k, 1:500)). Antibodies were diluted in blocking solution and sections were incubated for 24 hr. After several rinses in PBS/Tween, sections were incubated in secondary antibodies (Alexa goat anti-mouse 488 and Alexa goat anti-rabbit 568, Alexa Goat anti-rat 568, Invitrogen, 1:500) for 2 hr. Finally, sections were washed in PBS/Tween and mounted under ProLong Gold Antifade fluorescence media with DAPI (Invitrogen). Images of whole-brain sections were acquired with a slide scanner using the Aperio Scanscope FL Scanner, (Leica Biosystems Inc.). The slide scanner uses a monochrome TDI line-scan camera, with a PC-controlled mercury light source to capture high-resolution, seamless digital fluorescent images. Images of hippocampi were acquired on a Zeiss 780 meta confocal microscope using a 40 × oil immersion lens. Counts were made of cells expressing the Cre-dependent tdTomato fluorescent reporter. Five *Amigo2*-icreERT2+; *ROSA*-tdTomato± mice were used for this analysis with 3–5 50-μm sections per animal spanning the anterior-posterior extent of CA2. Sections were stained for PCP4 and colocalization of PCP4 with tdTomato was assessed in a total of 5248 cells.

## Neurophysiological data acquisition and behavioral tracking

Neural activity was transmitted via a 32-channel wireless 10× gain headstage (Triangle BioSystems International, Durham, NC) and was acquired using the Cerebus acquisition system (Blackrock Microsystems, Salt Lake City, UT). Continuous LFP data were band-pass filtered at 0.3–500 Hz and stored at 1,000 Hz. Single unit data were sampled at 30 kHz and high-pass filtered at 250 Hz. Neurophysiological recordings were referenced to a silver wire connected to a ground screw secured in the posterior parietal bone. To confirm that gamma power activity recorded in hippocampus and PFC were not artifacts of volume conduction or apparent brain-wide increases in low-gamma power due to differential voltage measurement between the active recording electrode and the ground screw, in some animals, one wire per bundle targeting hippocampus or PFC, was positioned either in the cortex above hippocampus or in the striatum lateral to PFC. Referencing signals to these short or lateral wires showed LFPs that increased or decreased in gamma power upon CNO administration to hM3Dq or hM4Di-infused mice, respectively, similar to recordings that were referenced to the ground screw. For behavioral tracking, the X and Y coordinates in space of a light-emitting diode

(for use with color camera) or a small piece of reflective tape (for use with infrared camera) present on the wireless headstage were sampled at 30 Hz using Neuromotive Software (Blackrock Microsystems) and position data were stored with the neural data.

For recordings from mice infused with hM3Dq AAV, baseline data was acquired for at least 20 min followed by treatment with vehicle (10% DMSO in saline) or CNO (0.05–4 mg/kg CNO dissolved in DMSO to 50 mM then suspended in saline, SQ) and recording continued for an additional 2 hr. During the entire recording time, mice were inside of an open-field arena, which was a custom-built, five-sided (open top) dark arena (approximately 80 cm long x 80 cm wide x 100 cm high). The walls and floor of the arena were constructed from black-colored Plexiglass. Mice administered various doses of CNO were first treated with vehicle and then increasing doses of CNO at three-day intervals. Room light remained illuminated, but a curtain was placed around the open field chamber during recordings. For neurophysiology experiments on hM4Di AAV-infused mice, after connecting headstages to the animals' electrodes, animals were administered either vehicle or CNO (5 mg/kg, SQ) then returned to their cage for 30 min before starting recording. Room lights were turned off and red lights were illuminated after administering vehicle or CNO. After 30 min had elapsed, mice were placed in the open field arena for recording. This behavioral procedure was employed in an effort to drive endogenous gamma activity by increasing locomotor activity (see *Figure 2—figure supplement 4E–F*). Gamma power measurements were made during periods when the animals were moving at $\geq 7$ cm/sec. Ripple measurements were made when the animals were moving $\leq 0.5$ cm/sec.

Following recordings, neurons for single-unit recordings were sorted into individual units by tetrode mode-based cluster analysis in feature space using Offline Sorter software (Plexon Inc., Dallas, TX). Autocorrelation and cross-correlation functions of spike times were used as separation tools. Only units with clear refractory periods and well-defined cluster boundaries were included in the analyses. Putative pyramidal cells and interneurons were distinguished based on autocorrelation plots (peak within 10 msec representing bursting), waveforms (broad waveforms, with a peak to valley spike width of >300 μsec) and mean firing rates (<5 Hz during baseline recording; *Skaggs et al., 1996*). Only pyramidal cells were included in analyses.

For simultaneous optogenetic stimulation and LFP recording, fiber optic probes (200 μm diameter) were connected to a Plexbright four-channel optogenetics controller through a wired tether, and Radiant software (Plexon, Inc.) was used to drive light stimulation. Electrophysiological recordings were made from awake, behaving mice during periods of run and rest (behavioral state not separated for these experiments) using 32-channel head stages digitized at 16-bit resolution and acquired at 40 kHz using the OmniPlex D Neural Data Acquisition System (Plexon, Inc.). Continuous neural data were low-pass filtered at 500 Hz and sampled at 1000 Hz. For these experiments, baseline recordings were obtained for several minutes before delivering fiber optic stimulation. Stimulation consisted of five pulses delivered at 10 Hz, with each pulse being five msec in duration and with a current intensity of 200 mA delivered to the light emitting diode. One train was delivered per minute for 3 min. Animals were then administered either vehicle or CNO (5 mg/kg, SQ), and LFP responses to light stimulation were made following identical stimulation parameters between 20 min and 24 hr following vehicle/CNO treatment. Response amplitudes were measured from evoked voltage deflections time locked to the optogenetic stimulation events.

## Electrode localization

Upon completion of electrophysiology studies, mice were perfused with 4% paraformaldehyde. Heads with electrodes remaining in place in brains were then submerged in 4% paraformaldehyde for 24 – 48 hr. Electrodes were carefully removed and brains were submerged in 30% sucrose/PBS and sectioned at 40 μm on a cryostat or vibratome.

## Electrophysiology data analysis

The experimenter was blind to the genotype of animals at the time of electrophysiology recordings and data analysis. All neuronal data analyses were performed using Neuroexplorer software (Nex Technologies, TX) and MATLAB (MathWorks, Inc., Natick, MA) with the Chronux toolbox for MATLAB (http://chronux.org/). Statistical analyses were performed using GraphPad Prism version 6.

Identical analyses were used for all hM3Dq and hM4Di spectral measures. All hippocampal LFP measures were derived from an electrode channel connected to a wire positioned in the cell body layer, as determined by the presence of ripples. Data collected during the 30 to 60 min time window following CNO administration were first divided into periods of running (>7 cm/s) or resting (<0.5 cm/s and limited to up to 20 s once an animal has started moving <0.5 cm/s). As expected, theta and high-gamma power were greater during periods running than periods of resting (*Figure 2—figure supplement 3*). Running and resting LFP subsets were then z-scored to control for changes in overall signal amplitude (and, consequently, power) over the course of up to 2 weeks of recordings (in the case of hM3Dq animals in which multiple doses of CNO or vehicle were administered every 2 to 3 days). LFPs were then filtered using a zero-phase offset filter in the theta (5 – 10 Hz; *Bragin et al., 1995*; *Harris et al., 2002*; *Igarashi et al., 2014*), low-gamma (30 – 60 Hz; *Tort et al., 2008*; *Tort et al., 2010*) or high-gamma (65 – 100 Hz; *Tort et al., 2008*; *Tort et al., 2010*) range. The Chronux function mtspectrum, a multitaper spectral estimate, was used with five tapers, and resulting spectral values were smoothed. For all treatments, spectral measures were made during each of run and rest periods during the 30 to 60 min following treatment. Spectral density plots for each behavioral state, each treatment and in each recording site were averaged across animals according to genotype and AAV infused. Plots shown in figures are measured in arbitrary units due to z-scoring of LFPs, as described above. Peak powers in each frequency range were collected to compare changes in peak theta, low-gamma or high-gamma power according to treatment. Cross frequency coupling of theta phase and low gamma (30 – 55 Hz) amplitude were also measured from hippocampal LFPs during periods of running following each treatment using the method of (*Tort et al., 2008*; *Tort et al., 2010*). Coherence measures were performed using the Chronux function cohgramc (*Bokil et al., 2010*), and mean low-gamma coherence was measured over the 30 – 60 Hz frequency range from the run and rest subsets described above. To rule out volume conduction as a source of any change in coherence, we measured phase lag of low-gamma filtered LFPs in CA2 and PFC and confirmed the absence of zero-phase lag in our recordings. Phase angles were obtained by convolution for CA2 and PFC signals. Phase angle differences were taken between the two signals, and v-tests were used to determine statistically significant difference from zero or 2π (*Cohen, 2014*). Phase angle differences were significantly different from both zero and 2π. Power and coherence values measured for each treatment were compared using appropriate statistical tests, listed in text, after data were checked for normal distributions and equal variance. Of note, as a consequence of z-scoring LFPs for periods of run and rest separately, theta power appears to be similar between the two states. However, measurement of spectral power without z-scoring LFPs for run and rest shows that theta power is higher during periods of running than during rest (see *Figures 2A*; 4 mg/kg spectrogram shows strong theta power during periods of running). We did not observe a difference in time spent running or in running speed during recordings for any of the treatments.

Ripple events were identified according to modified methods previously described (*Suh et al., 2013*; *Boehringer et al., 2017*) from recordings originating from the pyramidal cell layer during periods of rest in hM3Dq (eight animals with dorsal CA1 recordings) and hM4Di (two animals with dorsal CA1 recordings, four animals with intermediate CA1 recordings) animals. Z-scored signals were denoised with an IIR notch filter at 60 and 180 Hz and filtered between 100 and 300 Hz with a 69-order FIR zero phase shift filter. Signals were then Hilbert transformed, and the absolute value envelopes were smoothed with a 50-ms window. Envelope amplitude deflections that exceeded three standard deviations from the mean amplitude (i.e. mean +3 standard deviations) for more than 30 ms were counted as ripple events. Deflections within 200 ms of a previous ripple event were excluded. Ripple event frequency and ripple amplitude were measured and appropriate statistical tests were applied, as listed in the text, after data were checked for normal distributions and equal variance.

## Results reporting and data availability

For each experiment presented within the Results section and in figures, the number of replicates is presented as 'N' when indicating the number of animals that were used for the experiment or as 'n' when referring to the number of neurons used for the experiment. Statistical tests used for each experiment are presented in the text. Statistical significance was based on a p-value of 0.05. All error

bars in graphs represent standard error of the mean. The data used to generate bar graphs in figures are listed in *Supplemental file 1*.

## Acknowledgements

This research is supported by the Intramural Research Program of the U.S. NIH, National Institute of Environmental Health Sciences (Z01 ES100221 to SMD). We wish to thank Guohong Cui, Jesse Cushman and the members of the Dudek lab for critically reading this manuscript, as well as Jesse Cushman and the NIEHS Neurobehavioral Core, the Fluorescence Microscopy and Imaging Center, and the animal care staff at NIEHS for their support.

## Additional information

### Funding

| Funder | Grant reference number | Author |
|---|---|---|
| National Institute of Environmental Health Sciences | ES100221 | S M Dudek |

The funders had no role in study design, data collection and interpretation, or the decision to submit the work for publication.

### Author contributions

Georgia M Alexander, Conceptualization, Formal analysis, Investigation, Visualization, Methodology, Writing—original draft; Logan Y Brown, Validation, Investigation, Methodology, Writing—review and editing; Shannon Farris, Formal analysis, Investigation, Visualization, Methodology, Writing—review and editing; Daniel Lustberg, Caroline Pantazis, Investigation, Writing—review and editing; Bernd Gloss, Nicholas W Plummer, Investigation, Methodology; Patricia Jensen, Supervision, Methodology, Writing—review and editing; Serena M Dudek, Conceptualization, Resources, Supervision, Funding acquisition, Writing—original draft, Project administration, Writing—review and editing

### Author ORCIDs

Georgia M Alexander http://orcid.org/0000-0003-4245-4417
Shannon Farris http://orcid.org/0000-0003-4473-1684
Serena M Dudek http://orcid.org/0000-0003-4094-8368

### Ethics

Animal experimentation: This study was performed in strict accordance with the recommendations in the Guide for the Care and Use of Laboratory Animals of the National Institutes of Health. All of the animals were handled according to approved institutional animal care and use committee (ACUC) protocol (#2009-0023) of the NIEHS (A4149-1). All surgery was performed under ketamine and xylazine anesthesia, and every effort was made to minimize suffering.

### Decision letter and Author response

Decision letter https://doi.org/10.7554/eLife.38052.032
Author response https://doi.org/10.7554/eLife.38052.033

## Additional files

### Supplementary files

• Supplementary file 1.
DOI: https://doi.org/10.7554/eLife.38052.025

• Transparent reporting form
DOI: https://doi.org/10.7554/eLife.38052.026

## Data availability

The data used to generate bar graphs in figures are listed in Supplementary File 1.

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
