## [Decision Letter]

Thank you for submitting your article "CA2 Neuronal Activity Controls Hippocampal Low Gamma and Ripple Oscillations" for consideration by *eLife*. Your article has been reviewed by three peer reviewers, one of whom is a member of our Board of Reviewing Editors, and the evaluation has been overseen by Joshua Gold as the Senior Editor. The reviewers have opted to remain anonymous.

The reviewers have discussed the reviews with one another and the Reviewing Editor has drafted this decision to help you prepare a revised submission.

Summary:

This paper uses chemogenetic manipulations and electrophysiological recordings from freely behaving mice to investigate how activation or inactivation of CA2 affects hippocampal rhythms, PFC gamma rhythms, and CA1-PFC coherence. To achieve this, the authors developed a novel mouse that specifically targets pyramidal cells in CA2. The authors report that low frequency gamma rhythms are enhanced by CA2 activation, whereas sharp wave-ripples are enhanced by CA2 inactivation. These findings provide novel insights into gamma oscillators in hippocampus and also reveal the potentially important role of CA2 in hippocampal oscillations related to cognition and memory consolidation.

All reviewers were generally positive about the importance of these findings. However, all reviewers also agreed that there are a number of issues that require further attention.

Essential revisions:

1) A major concern is whether animals' behavior differed during CA2 manipulations compared to controls. One would expect to see increased low frequency gamma rhythms at higher running speeds in mice (Chen et al., 2011), and the authors should also note that this paper showed that the theta phase-gamma amplitude modulation index changes with running speed. The authors write, "We did not observe a difference in time spent running or in running speed during recordings for any of the treatments", but no results or statistics are provided to back up this statement. In Figure 2A, it seems as though the theta power band in the color plots is more prominent for the vehicle group, possibly suggesting that the CNO treatments affected animals' behavior in a way that could bias gamma power measurements. Related to this point, why were mice returned to their home cage prior to recordings for the hM4Di group but not for the hM3Dq group? This could cause a difference in general activity level or attentional state that could bias measurements of rhythmic activity.

2) Throughout the paper, the recording locations need to be clearly identified and emphasized. In the Results section related to Figure 2, it is unclear if the results arise from signals recorded in CA1, CA2, or both. It would be best to show CA1 and CA2 separately to best interpret results. In the hM3Dq mice, there are many recording sites close to CA2. Thus, one wonders how much of the increase in low gamma power is a rather trivial result of the increased firing of the CA2 cells. That is, does the gamma power in the LFP merely reflect the activity level of the local neuronal population? Also, while the placement of electrodes for the hM3Dq mice is near the border between CA2 and proximal CA1, the electrode placements for the hM4Di mice run the gamut from proximal to distal CA1, and also at different levels of the longitudinal axis. Although it is not obvious that this difference in placement can account for the different results of excitation of CA2 in the former set of mice and inhibition of CA2 in the latter set, this difference in electrode placement should be emphasized explicitly and prominently in the Results section so that readers are made aware of this issue. Also, the authors need to show the histological placement of electrodes for the control (Cre-) mice in Figure 2—figure supplement 1 and Figure 5—figure supplement 2, not just the Cre+ mice. It is crucial that the electrode placements be equivalent for Cre+ and Cre- mice in order for the conclusions to hold.

3) The authors present several arguments against volume conduction (e.g., presenting data from electrodes that missed their targets). However, it seems as though it would be more convincing to present single unit phase-locking of spikes. The authors mention tetrodes in the Materials and methods section (subsection “Electrode Implantation”), but no single unit analyses are presented (other than optogenetic manipulation of spike rate in one mouse). Was spiking within each region modulated by the slow gamma that increased with CNO administration? Alternatively, the authors could perform analyses specifically developed to address volume conduction.

4) Some of the statistical reporting in the paper is a bit strange, and the rationale for the way in which some of the results and statistics are presented is unclear. For gamma power modulation by theta phase, the authors report "a trend toward significantly decreasing with CNO treatment" and a p-value of 0.0612. Yet, in the subsection “Increasing CA2 pyramidal cell activity increases hippocampal and prefrontal cortical low gamma power”, they report a p-value of less than 0.0501 for high frequency gamma power and report that this is not significant. This type of reporting of results makes it seem as though the authors are choosing to focus on some results and ignore others, regardless of statistics. Also, is it possible that the lack of significance (p <.0.0501) for high gamma is the result of low statistical power? Also, throughout the paper, when making comparisons across groups, the authors' results would be stronger if they assessed all rhythm measurements together in the same ANOVA then checked for interaction effects. For example, if the authors want to claim that chemogenetic activation of CA2 enhances low frequency gamma but not high frequency gamma, one would expect to see a significant interaction between rhythm type and CNO treatment group, with a main effect of CNO treatment only observed for low frequency gamma. Also, it is unclear why a t-test was used to assess effects of hM4Di infusion (subsection “CA2 pyramidal cell inhibition decreases hippocampal and prefrontal cortical low gamma power”, second paragraph), but a repeated measures ANOVA was used to assess significance of hM3Dq results (fourth paragraph).

5) For hippocampal recordings, the authors measure "beta" activity. However, it is unclear what is actually being measured because beta rhythms are typically not prevalent in CA1 in tasks lacking olfactory sampling of stimuli. What is likely being measured is the theta harmonic, which can clearly be seen in some of the spectra (e.g., Figure 5A), especially considering that "beta" results seem to mirror theta results (e.g., Figure 2B-C).

6) The authors assess theta and high frequency gamma rhythms during immobility, but this seems strange. Theta should not occur during immobility. Immobility is thought to be a sharp wave-ripple-associated behavioral state. Low gamma rhythms have been reported to co-occur with sharp wave-ripples, but high frequency gamma rhythms have not. What are the authors actually measuring during rest when they are assessing power in the theta and high frequency gamma ranges?

7) The results in Figure 2 are confusing. In example data of Figure 2A, it's quite obvious that low gamma power increased during CNO treatment. In addition, CNO also induced increased power in the 60-80 Hz frequency band compared to the vehicle condition, particularly in running periods when theta was strong. Furthermore, there is a trend toward increasing high gamma as a function of CNO dosage in Figure 2D. It looks like high gamma increased for half of the animals but not for the other half. If the authors specifically examine 60-80Hz is there a significant increase in 60-80Hz power with higher CNO dosage? Is there something different about the exact recording location for the 3 animals that relates to an increase in high gamma? More broadly, was there any difference between running and resting state for 60-80 Hz gamma band in either CNO or vehicle groups? Something also seems strange about the theta results presented in Figure 2B. For the run data, why is one animal so different than the rest? What exactly is significant in the run data? There seems to be no effect apparent, just a lot of variability across animals.

---

## [Author Response]

Essential revisions:1) A major concern is whether animals' behavior differed during CA2 manipulations compared to controls. One would expect to see increased low frequency gamma rhythms at higher running speeds in mice (Chen et al., 2011), and the authors should also note that this paper showed that the theta phase-gamma amplitude modulation index changes with running speed. The authors write, "We did not observe a difference in time spent running or in running speed during recordings for any of the treatments", but no results or statistics are provided to back up this statement. In Figure 2A, it seems as though the theta power band in the color plots is more prominent for the vehicle group, possibly suggesting that the CNO treatments affected animals' behavior in a way that could bias gamma power measurements. Related to this point, why were mice returned to their home cage prior to recordings for the hM4Di group but not for the hM3Dq group? This could cause a difference in general activity level or attentional state that could bias measurements of rhythmic activity.

To address whether the animals’ behavior differed during CA2 manipulations, we compared time spent running and run velocity during periods of LFP sampling following either vehicle or CNO for both hM3Dq and hM4Di animals. These data are now included in the revised manuscript, are presented as Figure 2—figure supplement 4. Among hM3Dq mice, we observed no significant difference in percent of time running for Cre+ or Cre- mice across treatments (Figure 2—figure supplement 4A). We also found no significant difference in running velocity for Cre+ or Cre- mice across treatments (Figure 2—figure supplement 4B). Similarly, among hM4Di mice, we observed no significant difference in percent of time running for Cre+ or Cre- mice between treatments (Figure 2—figure supplement 4C). We also found no significant difference in running velocity for Cre+ or Cre- mice between treatments (Figure 2—figure supplement 4D). From these analyses, we conclude that CNO treatment did not significantly impact locomotion for either hM3Dq or hM4Di mice.

Regarding the theta phase-gamma amplitude relationship with running speed, the apparent decrease in modulation index following CNO treatment among hM3Dq animals likely reflects the increased occurrence of gamma cycles following CNO treatment, which is seen during the expected descending phase of the theta cycle and spreads throughout the theta cycle at higher doses of CNO. Therefore, the decreased modulation index with CNO would not necessarily reflect decreased running speed, which is supported by the lack of significant difference in running speed or percent of time running with CNO in hM3Dq animals discussed above.

As the reviewer(s) note, we returned mice to their home cage prior to recordings for the hM4Di group but not for the hM3Dq group. Our objective for the hM4Di experiments was to drive endogenous gamma oscillations and then ask whether silencing CA2 activity inhibited endogenous gamma oscillations. We compared percent of time running and run velocity following vehicle treatment for hM3Dq mice and hM4Di mice (Cre+ and Cre- combined) and found that both time spent running and run velocity were significantly greater in the hM4Di mice than hM3Dq mice (Figure 2—figure supplement 4E-F), which likely reflected the different procedures described above, as hoped.

Regarding the color plots in Figure 2A appearing to show a stronger theta band during the vehicle treatment, the relative color difference between spectral plots representing LFPs collected during experiments with various treatments likely reflects variability of recordings over the days between successive treatments. Importantly, the theta band does not appear to change in color from before vehicle/CNO to after vehicle/CNO treatment, further supporting the conclusion that vehicle/CNO treatment did not impact animals’ locomotor activity.

2) Throughout the paper, the recording locations need to be clearly identified and emphasized. In the Results section related to Figure 2, it is unclear if the results arise from signals recorded in CA1, CA2, or both. It would be best to show CA1 and CA2 separately to best interpret results. In the hM3Dq mice, there are many recording sites close to CA2. Thus, one wonders how much of the increase in low gamma power is a rather trivial result of the increased firing of the CA2 cells. That is, does the gamma power in the LFP merely reflect the activity level of the local neuronal population?

In the revised manuscript, we have divided the hM3Dq data into recordings made from CA2, proximal CA1 and intermediate CA1, as determined by electrode placement images (Figure 2—figure supplement 1). These new analyses are presented in Figure 2—figure supplement 6. We found no significant difference in the magnitude of low gamma power change between recordings from close to CA2 and further from CA2 during periods of run (Figure 2—figure supplement 6A-B) or periods of rest (Figure 2—figure supplement 6C-D). These data do not support the conclusion that the increase in low gamma power simply reflects an increase in firing rate of CA2 pyramidal cells. We now emphasize the electrode positioning within the hM3Dq group (subsection “Increasing CA2 pyramidal cell activity increases hippocampal and prefrontal cortical low gamma power”, seventh paragraph), and present these analyses showing no difference in gamma power increase according to recording location within hippocampus.

Also, while the placement of electrodes for the hM3Dq mice is near the border between CA2 and proximal CA1, the electrode placements for the hM4Di mice run the gamut from proximal to distal CA1, and also at different levels of the longitudinal axis. Although it is not obvious that this difference in placement can account for the different results of excitation of CA2 in the former set of mice and inhibition of CA2 in the latter set, this difference in electrode placement should be emphasized explicitly and prominently in the Results section so that readers are made aware of this issue.

As discussed in the manuscript, we chose to focus our hM4Di analyses on the target area of CA2 neurons because the primary effect of hM4Di is on neurotransmitter release rather than somatic membrane potential, and thus spike activity. CA2 neurons project to CA1 in the same anterio-posterior plane as well as toward intermediate to posterior CA1. Therefore, we measured effects of hM4Di in dorsal and intermediate/posterior CA1. That said, in our revised manuscript, we have divided the hM4Di animals into those with recordings in dorsal CA1 and those with recordings in intermediate CA1. These analyses are now presented in Figure 5—figure supplement 3. Statistical analyses showed a main effect of drug treatment during periods of running (Figure 5—figure supplement 3A-B) but no main effect of recording location (dorsal vs. intermediate CA1) and no interaction effect. During periods of rest, we found no significant effect of recording location, drug treatment or an interaction (Figure 5—figure supplement 3C-D). This analysis is thoroughly described in the Results section of the main text (subsection “CA2 pyramidal cell inhibition decreases hippocampal and prefrontal cortical low gamma power”, second paragraph) to clearly articulate the presence of two target recording sites included in the CA1 hM4Di dataset.

Also, the authors need to show the histological placement of electrodes for the control (Cre-) mice in Figure 2—figure supplement 1 and Figure 5—figure supplement 2, not just the Cre+ mice. It is crucial that the electrode placements be equivalent for Cre+ and Cre- mice in order for the conclusions to hold.

Histological electrode placements for Cre- animals are now shown in the associated supplementary figures (now Figure 2—figure supplement 1 and Figure 5—figure supplement 2).

3) The authors present several arguments against volume conduction (e.g., presenting data from electrodes that missed their targets). However, it seems as though it would be more convincing to present single unit phase-locking of spikes. The authors mention tetrodes in the Materials and methods section (subsection “Electrode Implantation”), but no single unit analyses are presented (other than optogenetic manipulation of spike rate in one mouse). Was spiking within each region modulated by the slow gamma that increased with CNO administration? Alternatively, the authors could perform analyses specifically developed to address volume conduction.

We recorded single unit activity from CA2 to demonstrate an increase in CA2 pyramidal cell firing rate by hM3Dq upon CNO administration and indeed found increased firing rate of these neurons with CNO (shown in Figure 2—figure supplement 2). We did not record single unit firing from areas outside of CA2, including CA1 and PFC. To address volume conduction, we measured phase lag of low gamma filtered LFPs in CA2 and PFC and confirmed the absence of zero-phase lag in our recordings. Phase angles were obtained by convolution for CA2 and PFC low gamma filtered signals. Phase angle differences were taken between the two signals, and v-tests were used to determine statistically significant difference from zero or 2π^1^.

Indeed, phase angle differences were significantly different from both zero and 2π. This information is now provided in the Materials and methods subsection “Electrophysiology Data Analysis”.

4) Some of the statistical reporting in the paper is a bit strange, and the rationale for the way in which some of the results and statistics are presented is unclear. For gamma power modulation by theta phase, the authors report "a trend toward significantly decreasing with CNO treatment" and a p-value of 0.0612. Yet, in the subsection “Increasing CA2 pyramidal cell activity increases hippocampal and prefrontal cortical low gamma power”, they report a p-value of less than 0.0501 for high frequency gamma power and report that this is not significant. This type of reporting of results makes it seem as though the authors are choosing to focus on some results and ignore others, regardless of statistics. Also, is it possible that the lack of significance (p <.0.0501) for high gamma is the result of low statistical power?

We appreciate this comment and have now adjusted the wording to be clear that *p* values less than 0.05 are not considered to be significant. It is possible that with more animals, high gamma power would also be significantly affected by CA2 activity modulation. If further animals were to yield a significant effect on high gamma power, this would most likely be seen in the hM3Dq mice. In that hypothetical situation, one possible mechanism that could account for the increase in high gamma power could be through CA2 activity driving activity in entorhinal cortex via CA1, which then could feed back onto CA1 and promote high gamma activity. However, using the opposite manipulation – decreasing CA2 activity with hM4Di – we found a selective decrease in low gamma power upon CNO administration, and not high gamma power, leading us to conclude that CA2 activity is needed for low gamma oscillations in CA1 but not necessarily high gamma oscillations.

Also, throughout the paper, when making comparisons across groups, the authors' results would be stronger if they assessed all rhythm measurements together in the same ANOVA then checked for interaction effects. For example, if the authors want to claim that chemogenetic activation of CA2 enhances low frequency gamma but not high frequency gamma, one would expect to see a significant interaction between rhythm type and CNO treatment group, with a main effect of CNO treatment only observed for low frequency gamma. Also, it is unclear why a t-test was used to assess effects of hM4Di infusion (subsection “CA2 pyramidal cell inhibition decreases hippocampal and prefrontal cortical low gamma power”, second paragraph), but a repeated measures ANOVA was used to assess significance of hM3Dq results (fourth paragraph).

Repeated-measures ANOVAs were used to compare hM3Dq data because multiple doses of CNO were studied (vehicle, 0.5 mg/kg, 1 mg/kg, 2 mg/kg and 4 mg/kg CNO). We studied multiple doses of CNO in the hM3Dq mice to take advantage of the dose dependent nature of CNO effects on hM3Dq and, consequently, neuronal activity. Ttests were used for hM4Di animals because only two treatments were being compared (vehicle and 5 mg/kg CNO). We did not study dose-dependent effects of CNO in the hM4Di experiments but rather aimed to use a high enough dose of CNO to inhibit CA2 output to the maximum allowable extent afforded by the DREADD ligand.

As the reviewer requested though, in the revised manuscript, we have also performed two-way ANOVAs for the both hM3Dq and hM4Di groups with the two factors of treatment and gamma range. We thank the reviewer for this suggestion, as we did indeed find interaction effects between treatment and gamma range. These analyses are now presented in Figure 2—figure supplement 5 and Figure 5—figure supplement 5.

For the hM3Dq group, for periods of running, we found a main effect of treatment, a main effect of gamma range and a significant interaction between treatment and gamma range (Figure 2—figure supplement 5A). Follow-up post hoc tests revealed a significant increase in low gamma for 0.5, 1, 2, and 4 mg/kg CNO. Post hoc tests did not reveal a significant change in high gamma power for any dose of CNO. For periods of rest, we found a significant main effect of gamma range, a main effect of treatment, and a significant interaction between gamma range and treatment (Figure 2—figure supplement 5B). Follow-up post hoc tests revealed a significant increase in low gamma power for 0.5, 1, 2, and 4 mg/kg CNO. Post hoc tests did not reveal a significant change in high gamma power for any dose of CNO. Among Cre- animals, during periods of running, we found a significant main effect of gamma range but no main effect of treatment and no interaction (Figure 2—figure supplement 5C). Similarly, in Cre- mice during periods or rest, we found a main effect of gamma range but no effect of treatment or an interaction (Figure 2—figure supplement 5D).

For the hM4Di group, during periods of running, we found a main effect of treatment, a main effect of gamma range and a significant interaction between treatment and gamma range (Figure 5—figure supplement 5A). Follow-up post hoc tests revealed a significant decrease in low gamma power with CNO treatment but no significant change in high gamma power. For periods of rest, we found a significant main effect of gamma range but no main effect of treatment and no interaction between gamma range and treatment (Figure 5—figure supplement 5B).

Among Cre- animals, for the hM4Di group, during periods of running, we found a

significant main effect of gamma range but no main effect of treatment and no interaction (Figure 5—figure supplement 5C). Similarly, during periods or rest, we found a main effect of gamma range but no effect of treatment or an interaction (Figure 5—figure supplement 5D).

5) For hippocampal recordings, the authors measure "beta" activity. However, it is unclear what is actually being measured because beta rhythms are typically not prevalent in CA1 in tasks lacking olfactory sampling of stimuli. What is likely being measured is the theta harmonic, which can clearly be seen in some of the spectra (e.g., Figure 5A), especially considering that "beta" results seem to mirror theta results (e.g., Figure 2B-C).

The reviewer(s) raise a good point, and we have now removed these findings on beta oscillations from the manuscript.

6) The authors assess theta and high frequency gamma rhythms during immobility, but this seems strange. Theta should not occur during immobility. Immobility is thought to be a sharp wave-ripple-associated behavioral state. Low gamma rhythms have been reported to co-occur with sharp wave-ripples, but high frequency gamma rhythms have not. What are the authors actually measuring during rest when they are assessing power in the theta and high frequency gamma ranges?

The reviewer(s) bring up an interesting issue. Indeed, type I theta oscillations occur significantly more during mobility than during immobility, although a second type of theta, type II theta, occurring during aroused immobility have also been observed^2^. Thus, we think the oscillations in the theta frequency range that we report here likely reflect this type II theta. Consistent with this possibility, type I theta has a somewhat higher frequency (6-12 Hz) than type II theta (5-9 Hz), and in our dataset the peak theta frequency during running was significantly higher than the peak theta frequency during resting (for example, for hM3Dq Cre+ animals, main effect of behavioral state: F(1, 7)=20.42, *p* = 0.0027; main effect of treatment: F(4, 28) = 0.352, *p* = 0.8404; interaction: F(4, 28) = 0.7871, *p* = 0.5433; two-way ANOVA; post hoc tests of frequency during run vs. rest: vehicle: *p* = 0.0003, 0.5 mg/kg: *p* < 0.0001, 1 mg/kg: *p* = 0.0007, 2 mg/kg: *p* = 0.0002, 4 mg/kg: *p* = 0.0027, Bonferroni multiple comparisons test; Author response image 1). Of note though, hM3Dq mediated activation of CA2 did not result in a significant shift in peak theta frequency.

Regarding high gamma oscillations, high gamma power has been shown to increase with increasing running speed, but at low running speeds, high gamma power does not appear to be completely absent, particularly in CA3^3,4^. For the sake of providing information on each relevant hippocampal oscillation with a balanced presentation of running and resting data, we have presented high gamma oscillation power for both running and resting, although high gamma power is notably lower during periods of rest (see Figure 2—figure supplement 3).

7) The results in Figure 2 are confusing. In example data of Figure 2A, it's quite obvious that low gamma power increased during CNO treatment. In addition, CNO also induced increased power in the 60-80 Hz frequency band compared to the vehicle condition, particularly in running periods when theta was strong. Furthermore, there is a trend toward increasing high gamma as a function of CNO dosage in Figure 2D. It looks like high gamma increased for half of the animals but not for the other half. If the authors specifically examine 60-80Hz is there a significant increase in 60-80Hz power with higher CNO dosage? Is there something different about the exact recording location for the 3 animals that relates to an increase in high gamma?

We chose to use 65-80 Hz for the high gamma sampling range to minimize any possible contamination of the signal by 60 Hz line noise, although line noise has not been observed in our recordings. As requested by the reviewer, we have sampled the 60-80 Hz frequency band and asked whether high gamma power in that range was significantly increased with CNO treatment. We focused on hM3Dq animals during periods of run, as the reviewer notes particularly increased high gamma power in the 60-80 Hz range during periods of running. We found that power was not significantly increased by CNO in the 60-80 Hz frequency band (F(1.462, 10.23)=2.642, *p* = 0.1275, repeated-measures one-way ANOVA with Geisser-Greenhouse correction for unequal variance). These results are shown in Author response image 2.

**Author response image 2. respfig2:** 

Regarding the three animals with the greatest increase in high gamma power, these three animals had electrodes positioned in CA2 and in proximal CA1. The specific histology images matching the three animals with the greatest increase in high gamma power are shown below for the reviewers. There does not appear to be a correlation between the increase in high gamma power and a specific recording site within this sample population. As a further analysis to address the reviewer’s question, we also divided animals into the various areas of electrode implantation and analyzed high gamma power during running. The results show no main effect of recording location, CNO treatment or an interaction (main effect of location: F(2, 5) = 0.3812, *p* = 0.7013; main effect of treatment: F(4, 20)=1.553, *p* = 0.2255; interaction: F(8, 20) = 0.3717, *p* = 0.9234, two-way ANOVA; see Author response image 3).

**Author response image 3. respfig3:** 

More broadly, was there any difference between running and resting state for 60-80 Hz gamma band in either CNO or vehicle groups?

As demonstrated in the spectrograms in Figure 2A, there was greater high gamma power during running than resting. Due to the LFP normalization within each of the run and rest datasets, the power values for high gamma do not appear to differ significantly; however, to address the reviewer’s question, we analyzed non-normalized data for periods of run vs. periods of rest following vehicle treatment in hM3Dq animals. We found that high gamma power in the 60-80 Hz range was significantly greater during periods of running than during periods of resting, as expected (N=8, t(7)=4.996, *p* = 0.0016, paired t-test; Author response image 4). As the reviewer brings up an important point worth clarifying for the reader, we have now included a supplementary figure in the revised manuscript comparing theta and high gamma power from non-normalized LFPs during periods of running versus periods of resting in hM3Dq animals following vehicle treatment. For the supplementary figure, we used the high gamma frequency range of 65-80 Hz to be consistent with other high gamma data presented in the manuscript. In analyzing theta power and high gamma power between running and resting in hM3Dq animals following vehicle treatment, we found that both theta and high gamma power were significantly increased during periods of running (Figure 2—figure supplement 3A-C). We also compared theta frequency and found that peak theta frequency was significantly higher during periods of running than during resting (Figure 2—figure supplement 3D), consistent with the presence of type II theta during rest, as discussed in point #6 above.

**Author response image 4. respfig4:** 

Something also seems strange about the theta results presented in Figure 2B. For the run data, why is one animal so different than the rest? What exactly is significant in the run data? There seems to be no effect apparent, just a lot of variability across animals.

Indeed, one animal showed prominent theta power during running. We have confirmed the accuracy of the analysis and results for this animal and conclude that biological variability is responsible for the high theta power in this animal. Regarding the statistics for theta power results presented in Figure 2B, we compared theta power across treatments (vehicle, 0.5 mg/kg, 1 mg/kg, 2 mg/kg, or 4 mg/kg CNO) using the Friedman test, a non-parametric form of one-way ANOVA. We used the Friedman test due to the non-Gaussian distribution of data, as exemplified by the one animal showing higher theta power than the other animals in the cohort. The Friedman test reported a Friedman statistic of 11.3 with a *p* value of 0.0234. The Dunn’s multiple comparisons test of each CNO dose vs. vehicle reported the following adjusted *p*-values: 0.5 mg/kg, *p* = 0.3280; 1 mg/kg, *p* > 0.9999; 2 mg/kg, *p* > 0.9999; 4 mg/kg, *p* = 0.0708. As stated in the figure legend, we conclude that theta power varied significantly upon treatment. This conclusion is supported by the data, reporting the result of an appropriately chosen statistic based on non-Gaussian distribution while not overstating the data.

References:

1) Cohen, M. X. Analyzing Neural Time Series Data. (MIT Press, 2014).

Sainsbury, R. S. and Montoya, C. P. The relationship between type 2 theta and behavior. Physiol. Behav. 33, 621–626 (1984).

2) Sainsbury, R. S. & Montoya, C. P. The relationship between type 2 theta and

behavior. Physiol. Behav. 33, 621–626 (1984).

3) Zheng, C., Bieri, K. W., Trettel, S. G. and Colgin, L. L. The relationship between gamma frequency and running speed differs for slow and fast gamma rhythms in freely behaving rats. Hippocampus 25, 924–938 (2015).

4) Ahmed, O. J. and Mehta, M. R. Running speed alters the frequency of hippocampal gamma oscillations. Journal of Neuroscience 32, 7373–7383 (2012).

5) Fisahn, A., Pike, F. G., Buhl, E. H. and Paulsen, O. Cholinergic induction of network oscillations at 40 Hz in the hippocampus in vitro. Nature 394, 186–189 (1998).

6) Tamamaki, N., Abe, K. & Nojyo, Y. Three-dimensional analysis of the whole axonal arbors originating from single CA2 pyramidal neurons in the rat hippocampus with the aid of a computer graphic technique. Brain Res. 452, 255–272 (1988).

7) Cui, Z., Gerfen, C. R. & Young, W. S. Hypothalamic and other connections with

dorsal CA2 area of the mouse hippocampus. J. Comp. Neurol. 521, 1844–1866

(2013).